

# Including Effects of Watershed Heterogeneity in the Curve Number Method Using Variable Initial Abstraction

Vijay P. Santikari, Lawrence C. Murdoch

Department of Environmental Engineering and Earth Sciences, Clemson University, Clemson, SC 29634, USA

*Correspondence to*: Vijay P. Santikari (vsantik@g.clemson.edu)

**Abstract**

The curve number (CN) method was developed more than half a century ago and is still used in many watershed/water quality
models to estimate direct runoff from a rainfall event. Despite its popularity, the method is plagued by a conceptual problem
where CN is assumed to be constant for a given set of watershed conditions, but many field observations show that CN
decreases with event rainfall ($P$). Recent studies indicate that heterogeneity within the watershed is the cause of this behavior,
but the governing mechanism remains poorly understood. This study shows that heterogeneity in initial abstraction, $I_a$, can be
used to explain how CN varies with $P$. By conventional definition, $I_a$ is equal to the cumulative rainfall before the onset of
runoff, and is assumed to be constant for a given set of watershed conditions. Our analysis shows that the total storage in $I_a$
($I_{aT}$) is constant, but the effective $I_a$ varies with $P$, and is equal to the filled portion of $I_{aT}$, which we call $I_{aF}$. CN calculated
using $I_{aF}$ varies with $P$ similar to published field observations. This motivated modifications to the CN method, called Variable
$I_a$ Models (VIMs), which replace $I_a$ with $I_{aF}$. VIMs were evaluated against Conventional Models CM0.2 ($\lambda = 0.2$) and CM$\lambda$
(calibrated $\lambda$) in their ability to predict runoff data generated using a distributed parameter CN model. The performance of
CM0.2 was the poorest whereas those of the VIMs were the best in predicting overall runoff and watershed heterogeneity.
VIMs also predicted the runoff from smaller events better than the CMs, and eliminated the false prediction of zero-runoffs,
which is a common shortcoming of the CMs. We conclude that including variable $I_a$ accounts for heterogeneity and improves
the performance of the CN method while retaining its simplicity.

## 1. Introduction

The estimation of runoff from a rainfall event is of primary importance in applied hydrology. It is necessary in the engineering
design of small structures, post-event appraisals, environmental impact work, and other applications (Hawkins, 1993). One of
the most popular techniques used for this purpose is the Curve Number method, which has been in use for more than half a
century (Soil Conservation Service, 1956). The method uses a parameter called Curve Number (CN), which is assumed to
depend mainly on land cover, soil types, and antecedent conditions within a watershed.




Curve Number varies spatially due to watershed heterogeneity, and temporally due to changes in soil moisture, land cover, temperature, and other processes (Hawkins et al., 2008; Ponce and Hawkins, 1996; Rallison and Miller, 1982). CN also varies with the magnitude and spatiotemporal distribution of rainfall. When heterogeneity is known at sufficient detail, CN variation can be accounted by using a distributed parameter model. Otherwise this approach can introduce more parameters than can be reliably estimated from the available data, and cause large uncertainties in the predicted runoff. There are several ways to account for temporal variation of CN, each with its own advantages and shortcomings (Santikari and Murdoch, 2018). CN variation with the distribution of rainfall is usually ignored. CN method is most commonly applied as an event-scale lumped-parameter model, which is simple but also limited in its ability to account for the variations of CN. This diminishes the accuracy of its runoff predictions.

The objective of this work is to improve the event-scale lumped-parameter application of the CN method by describing an approach for incorporating the spatiotemporal variations of CN. The investigation is described in two papers. In this paper, effects of spatial variation of CN (heterogeneity) at the watershed scale are analyzed. Insights gained from this analysis are used to create modified models that account for heterogeneity. The modified models are evaluated using the runoff generated by a distributed parameter model applied to a hypothetical heterogeneous watershed. In a companion paper (Santikari and Murdoch, 2018), the modified models are refined by including an approach that accounts for the temporal variation of CN using antecedent moisture. The refined models, which account for spatial and temporal variability, are then evaluated using data from real watersheds.

## 1.1. Background

The CN method assumes that a rainfall event produces runoff ($Q$) when the event rainfall ($P$) exceeds the initial abstraction ($I_a$). $I_a$ includes interception storage (by tree canopy, roof tops and such), early infiltration, and surface depression storage. The effective rainfall, $P - I_a$, is partitioned between $Q$ and further infiltration ($F$). This is given by mass balance as

$$P - I_a = F + Q \quad \forall \quad P \geq I_a \tag{1}$$

Both $F$ and $Q$ are zero when $P \leq I_a$, and both increase with $P$ when $P > I_a$. It is assumed that $F$ has an upper limit, which is referred to as the potential maximum retention ($S$). In other words, $S$ is the total storage available for infiltration after the runoff begins.

The conceptual basis that defines the curve number method comes from the following assumption (Hawkins et al., 2008; NRCS, 2003; Ponce and Hawkins, 1996; Rallison and Miller, 1982; Woodward et al., 2002):

$$\frac{Q}{P - I_a} = \frac{F}{S} \tag{2}$$




i.e. the runoff coefficient (left hand side) is equal to the fraction of storage filled in $S$ (right hand side). Equation (2) is developed using the reasoning that the equality holds at the end points ($P \leq I_a$ and $P \rightarrow \infty$) (Hawkins et al., 2008; Rallison and Miller, 1982; Woodward et al., 2002), and that the behavior of both ratios in the intermediate range is essentially the same (Figure 1). When $P \leq I_a$, both $Q$ and $F$ are zero and therefore the ratios on either side of eq. (2) are zero. When $P > I_a$, both the ratios increase with $P$, whereas their rate of increase diminishes. At the limit of $P \rightarrow \infty$, both the ratios approach unity.

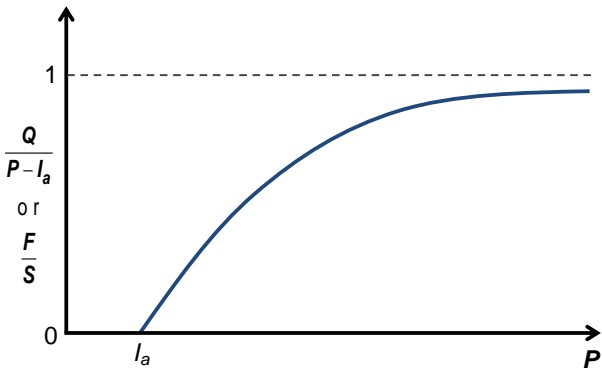

**Figure 1.** Presumed variation of the ratios in eq. (2) with event rainfall ($P$). $Q$ is event runoff, $I_a$ is initial abstraction, $F$ is cumulative infiltration after runoff begins, and $S$ is potential maximum retention (modified from Rallison and Miller (1982) Figure 2).

To eliminate the need for an independent estimation of $I_a$ (Ponce and Hawkins, 1996; Rallison and Miller, 1982), it is assumed that

$$I_a = \lambda S \tag{3}$$

where $\lambda$ is a dimensionless parameter called the initial abstraction ratio. Early field data suggested an optimum value of $\lambda = 0.2$ (Soil Conservation Service, 1956). However, more recent studies (Hawkins et al., 2008; Woodward et al., 2003) suggest that $\lambda = 0.05$ is more appropriate. Using eqs. (1), (2), and (3), $I_a$ and $F$ can be eliminated to give

$$Q = 0 \quad \forall \quad P \leq \lambda S$$

$$\tag{4}$$

$$Q = \frac{[P - \lambda S]^2}{P + (1 - \lambda)S} \quad \forall \quad P > \lambda S$$

Since the value of $\lambda$ is usually fixed (at 0.2 or 0.05), eq. (4) requires only one parameter, $S$, which varies within the range $0 \leq S \leq \infty$.



For convenience (Hawkins et al., 2008; Ponce and Hawkins, 1996), $S$ (units in mm) is mapped on to a dimensionless parameter called the Curve Number (CN) as

$$CN = \frac{25400}{254 + S} \tag{5}$$

so that CN is 100 when $S$ is zero, but approaches zero as $S$ approaches infinity. In practice, when $\lambda = 0.2$, CN ranges from around 30 (for vegetated surfaces with highly permeable soils) to close to 100 (for impermeable surfaces or soils) (USDA, 1986). Tabulated CN values for various land uses, soil types, and management scenarios are available in handbooks and manuals (NRCS, 2003; USDA, 1986). If a watershed has multiple land uses or soil types, typically, the CN is areally averaged. CN can also be determined from field data by solving eq. (4) for $S$ as

$$S = \frac{1}{2\lambda^2}\left[ 2\lambda P + (1-\lambda)Q - \sqrt{(1-\lambda)^2 Q^2 + 4\lambda PQ} \right] \tag{6}$$

and then using eq. (5). Conversely, when the CN of a watershed is known, $Q$ can be estimated for a rainfall event using eqs. (4) and (5).

The curve number method is appealing because it is based on an intuitive concept [eq. (2)], relies on only one parameter, has a large body of literature (Hawkins et al., 2008), and a comprehensive database (NRCS, 2003; USDA, 1986). It has been included in many watershed/water quality models such as SWAT (Soil and Water Assessment Tool) (Neitsch S.L. et al., 2005), CREAMS (Chemicals, Runoff and Erosion from Agricultural Management Systems), GLEAMS (Groundwater Loading Effects of Agricultural Management Systems) (Knisel and Douglas-Mankin, 2012), AnnAGNPS (Annualized Agricultural Non-point Source Pollution Model) (Bingner et al., 2011), EPIC (Environmental Policy Integrated Climate), APEX (Agricultural Policy/Environmental Extender) (Wang et al., 2012), and HydroCAD (HydroCAD, 2015). A physically-based modeling framework, such as the diffusive-wave approximation for overland flow coupled with the Richard's equation for unsaturated subsurface flow, e.g. (Panday and Huyakorn, 2004), may improve accuracy and resolution of model predictions compared to the CN method, when the necessary input data, expertise, and computing resources are available. However, the CN method will likely remain popular for many applications in runoff modeling because of its ease of use, wide knowledge base, and less demand on computational resources than many physically-based models.

**1.2. CN Variation with $P$**

Curve Number is assumed to be a watershed property that depends on the current conditions, but it also varies with $P$ [e.g. Figure 2(a) and 2(b)]. Hawkins (1993) and D'Asaro and Grillone (2012) evaluated approximately 100 watersheds in a wide range of settings and found the CN variation with $P$ to be common. In 75% of the watersheds they observed, CN decreased with increasing $P$ and asymptotically approached a constant value. Hawkins (1993) referred to this as *standard behavior*. In





20% of the watersheds, CN decreased with *P* but an asymptote was not attained within the range of the observed *P*. This was referred to as *complacent behavior*. In about 5% of the watersheds, the CN increased with *P* and asymptotically approached an apparent constant value. This behavior, referred to as *violent*, was often preceded by *complacent behavior* at smaller rainfalls. Hawkins (1993) hypothesized that the inverse relationship between CN and *P* may be due to some spurious

correlation between them, or due to a bias that inherently results from the selective omission of data from small storm events that failed to produce runoff. The reasoning is that large rainfalls always produce runoff but small rainfalls produce runoff only under wet conditions, when the CN is large. Therefore small CN values for small rainfalls go unrecorded.

In watersheds showing a *standard behavior*, CN was treated as an asymptotic function of *P* as

$$\mathrm{CN} = \mathrm{CN}_{\infty} + (100 - \mathrm{CN}_{\infty})\, e^{-kP} \tag{7}$$

where $\mathrm{CN} = \mathrm{CN}_{\infty}$ is the asymptote and *k* is a calibration parameter (Hawkins, 1993). $\mathrm{CN}_{\infty}$ is the smallest possible value of CN for a watershed and is approached only at large values of *P*. To develop eq. (7), measured values of *Q*, ideally for a large range of values of *P*, are needed. The usual procedure involves "frequency matching" the data (Hawkins, 1993), i.e. sorting the values of *P* and *Q* separately, and pairing them according to their rank. CN for each pair is then calculated using eqs. (5) and (6). Frequency matching reduces the scatter of data points around the best fit curve in a CN vs. *P* plot.

A *standard behavior* of CN was also observed in two watersheds (BC5 and BC1) near Greenville, South Carolina, USA [Figure 2(a) and 2(b)]. In these watersheds, CN (calculated using $\lambda = 0.2$) decreased from 97 to 50 as *P* increased from 2 mm to 128 mm. The data was characterized by a modest scatter ($\mathrm{R}^2 = 0.9$) about the best fit curve based on a quadratic function of *P*. Description of these watersheds is given by Santikari and Murdoch (2018). The justification for using quadratic functions follows from the analysis of heterogeneity presented in Section-2.

The approach used in Figure 2(a) and 2(b) avoids the commonly used frequency matching, e.g. (Hawkins, 1993). Each CN value in the plot was calculated using the *P-Q* pair from the same storm event. Frequency matching would significantly reduce the scatter in the plot, but it would also downplay the importance of CN variation due to antecedent conditions. Reducing the scatter by accounting for antecedent conditions, e.g. using antecedent moisture (Mishra et al., 2006), is a better approach.

The hypotheses given by Hawkins (1993) are valid, but insufficient to explain the *standard* and *complacent behaviors*. It may

be true that small rainfalls produce runoff only under wet (large CN) conditions and therefore only the large CN values are recorded. However, if one has a large enough sample of storms, some of the larger storms also must have occurred during wet conditions. For the larger storms, therefore, one would expect to see the whole spectrum of CN values ranging from the largest to the smallest. However, this is not the case. As *P* increases, the values of CN decrease consistently [Figure 2(a) and 2(b)].



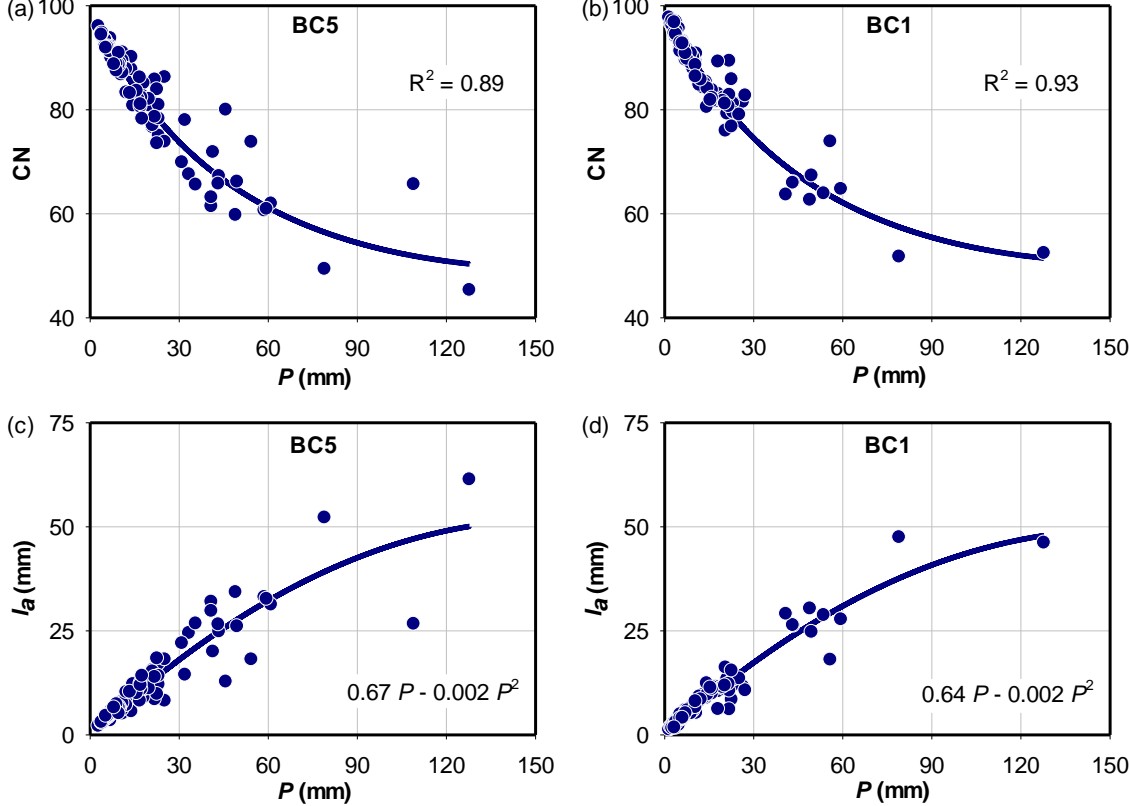

**Figure 2.** Variation of CN ($\lambda$ = 0.2) with $P$ in watersheds (a) BC5, (b) BC1, near Greenville, SC. Variation of $I_a$ with $P$ in (c) BC5, (d) BC1 (see Santikari and Murdoch (2018) for study area description). Best fit curves for $I_a$ are quadratic functions of $P$ with zero intercept. Corresponding best fit curves for CN were derived from those of $I_a$ using eqs. (3) and (5).

5    **1.3. Heterogeneity as a Cause of CN Variation with $P$**

Soulis and Valiantzas (2012) hypothesized that the observed variation of CN with $P$ in the *standard* and *complacent* cases is a consequence of watershed heterogeneity. They assumed a hypothetical heterogeneous watershed with two subareas characterized by different CNs. They then calculated the watershed runoff, for a range of values of $P$, as the area-weighted average of the runoffs from the subareas. Watershed CN calculated using this runoff varied with $P$ akin to the *standard*

10    *behavior*. The shape of the synthetically generated CN vs. $P$ curve could be matched with the observations by adjusting the areas of the subareas and their respective CNs. This idea can also be extended to multiple subareas so that the heterogeneity within a watershed can be represented more accurately. However, this could lead to problems of over-parameterization, non-uniqueness, and non-convergence as pointed out by Soulis and Valiantzas (2012).



In a later paper, Soulis and Valiantzas (2013) suggested using spatial information on land cover and soils to delineate the areal extent of subareas and constrain their respective CNs. This approach would reduce the number of calibrated parameters by half because it only requires the calibration of the CNs for the subareas. In essence, the multiple-subarea approach is similar to a distributed modeling approach that calculates the watershed runoff as the area-weighted average of the runoffs from the

subareas, e.g. SWAT (Gassman et al., 2007). The approach used by Soulis and Valiantzas (2013) attempts to match the observed and simulated values of CN, whereas that used by SWAT attempts to match the observed and simulated values of $Q$. Since CN and $Q$ are uniquely related for given values of $P$ and $\lambda$, these approaches are equivalent. A major implication of the work of Soulis and Valiantzas (2013) is that a distributed modeling approach can account for the *standard* and *complacent* *behaviors* of CN.

Using a single value of CN independent of $P$ in a heterogeneous watershed can cause a systematic error in $Q$, and lead to poor predictive ability of the method. This is because when CN is constant, $Q$ may be underestimated for small $P$ and overestimated for large $P$. This problem can be addressed either by treating CN as a function of $P$, e.g. asymptotic fitting (Hawkins, 1993), or by using a distributed modeling approach that accounts for heterogeneity in sufficient detail, e.g. SWAT (Gassman et al., 2007). An understanding of the mechanism of how watershed heterogeneity leads to the variation of CN with $P$ is also

important. It could help in accounting for this variation without resorting to fine discretization or over-parameterization of the CN method. To accomplish this, an analysis of the effect of heterogeneity on $I_a$ and $S$ is performed, which can then be used to understand the effect on CN.

## 2. Reevaluation of Initial Abstraction

The quantities CN, $I_a$ and $S$ are considered to be the properties of a watershed that depend on current conditions. In usual

practice, CN estimated for a certain set of conditions is applicable to any rainfall event occurring in those conditions irrespective of the magnitude of $P$. However, in every watershed evaluated by the previous studies (D'Asaro and Grillone, 2012; Hawkins, 1993) the CN varied with $P$. If so, since $I_a$ and $S$ are inversely related to CN [eqs. (3) and (5)], one can expect that they too vary with $P$ but inversely to that of CN. The calculated values of $I_a$ for watersheds BC5 and BC1 near Greenville, SC, increase with $P$ and appear to approach a constant at large values of $P$ [Figure 2(c) and 2(d)]. A plot of $S$ vs. $P$ would be

similar to the $I_a$ vs. $P$ plot, with the $y$–coordinate scaled by $1/\lambda$.

To evaluate the link between heterogeneity in $I_a$ and its variation with $P$, we looked at how the effective $I_a$ of a heterogeneous watershed is determined and whether it is affected by the magnitude of $P$. Our analysis shows that there is an inconsistency between the theoretical definition of $I_a$ and its calculated value at the watershed scale. It also shows how heterogeneity can cause $I_a$ to vary with $P$, and how this relates to variations of $S$ and CN with $P$.





## 2.1. Problems with the Current Usage of $I_a$

By the theoretical definition of $I_a$, if runoff is detected in the hydrograph, it is assumed that $I_a$ has been met for the watershed. Watersheds are heterogeneous combinations of various land use-soil-slope complexes. These are referred to as Hydrologic Response Units (HRUs) in SWAT (Gassman et al., 2007), and the same term is also used here. Each HRU is assumed to be

homogeneous, and is characterized by representative values of CN ($CN_i$) and $I_a$ ($I_{ai}$). During a rainfall event, the HRU with the smallest of the $I_{ai}$s will be the first to generate runoff. Assuming that this runoff reaches the watershed outlet, by definition, the $I_a$ of the watershed should be equal to the smallest of the $I_{ai}$s. This could even be zero if the watershed has surfaces such as open water bodies that cannot abstract the rainfall.

However, it is difficult to detect the exact moment of generation of runoff and determine the corresponding value of $I_a$, which

is equal to the cumulative precipitation at that moment. There have been studies (Shi et al., 2009; Woodward et al., 2003) that tried to determine $I_a$ from hydrographs. A problem with this approach is that there can be a time lag between runoff generation in headwaters and its detection at gauging station. Rainfall that occurs during this time lag is also included in $I_a$, leading to its overestimation. Another possible approach would be to collect observations from a large number of rainfall events and take $I_a$ to be equal to the smallest $P$ that produced runoff. This would eliminate the problem with the lag time, but $Q$ needs to be

insignificant to reduce the error in $I_a$. It should also be noted that $I_a$ determined this way is only representative of the antecedent conditions of the smallest event that produced runoff.

It may be difficult to measure $I_a$ directly, but it can be calculated for any event using eqs. (6) and (3). However, in medium to large rainfall events, even the HRUs with larger $I_{ai}$s will contribute to $Q$. Therefore, the calculated value of $I_a$ in these events will also be influenced by larger $I_{ai}$s. This value of $I_a$ tends to be greater than the smallest of the $I_{ai}$s. Moreover, it can be

expected to increase with $P$ as increasingly larger rainfalls generate runoff from HRUs with increasingly larger $I_{ai}$s. Thus, there is an inconsistency between the definition of $I_a$ and its calculated value at the watershed scale.

### 2.1.1. Spatial-scale effect on λ

Strictly adhering to the definition of $I_a$ at the watershed scale may also cause a spatial-scale effect on $\lambda$. Let us refer to the CN of the watershed as $CN_W$, and $I_a$ as $I_{aW}$. One of the common ways to determine $CN_W$ is to calculate it as the area-weighted

average of the $CN_i$s (NRCS, 2003) as

$$CN_w = \sum_{i=1}^{n} a_i \, CN_i \qquad (8)$$

where $a_i$ is the fractional area of the $i^{th}$ HRU. Note that the fractional areas must add up to unity. By definition, $I_{aW}$ is equal to the lowest of the $I_{ai}$s. Therefore, if $I_{a1} < I_{a2} < .... < I_{an}$ then





$$I_{aW} = I_{a1} \qquad (9)$$

From equations (3) and (5) it can be shown that CN and $I_a$ are related as

$$CN = \frac{25400}{254 + \left(\dfrac{I_a}{\lambda}\right)} \qquad (10)$$

If all the HRUs are assumed to have the same $\lambda = \lambda_i$, eqs. (8), (9) and (10) lead to

$$CN_1 > CN_2 > .... > CN_n$$
$$CN_W < CN_1$$
$$\frac{I_{aW}}{\lambda_W} > \frac{I_{a1}}{\lambda_i} \qquad (11)$$
$$\lambda_W < \lambda_i$$

where $\lambda_W$ is the effective initial abstraction ratio of the watershed. Therefore, if $\lambda$ is assumed to be the same among the component HRUs, it will have a smaller value at the watershed scale. This implies that $\lambda$ decreases with increasing spatial-scale. Therefore setting $\lambda$ constant, equal to 0.2 or 0.05, for all the spatial scales contradicts the definition of $I_a$. In any case, it is probably more accurate to calculate runoff at the HRU scale ($Q_i$) and take the area-weighted average of $Q_i$s, rather than take the area-weighted average of the $CN_i$s and calculate $Q$ at the watershed scale. It is also more appropriate because $Q$ is runoff per unit area whereas CN is a dimensionless index variable.

The inconsistencies in the usage of $I_a$ are a direct result of heterogeneity in a watershed. Moreover, heterogeneity also appears be responsible for the variation of $I_{aW}$ with $P$ [Figure 2(c) and 2(d)]. To verify this, a relationship between $I_{aW}$ and the magnitude and areal distribution of $I_{ai}$s needs to be developed.

## 2.2. $I_a$ in a Heterogeneous Watershed

Consider a watershed with four HRUs mainly characterized by their land use types viz. open waterbody ($I_{a0}$), urban area ($I_{a1}$), park ($I_{a2}$), and forest ($I_{a3}$) [Figure 3(a)], such that $I_{a0} = 0 < I_{a1} < I_{a2} < I_{a3}$. An open waterbody generates runoff during every rainfall event. Other land use types generate runoff depending on the magnitude of the rainfall, with land uses of larger $I_{ai}$ requiring larger magnitudes. The number of land use types contributing to the runoff, in other words the runoff contributing area, increases with rainfall. This process can be conceptualized by representing the storage distribution of $I_a$ as a series of bins where each bin corresponds to a HRU [Figure 3(b)]. The height and the width of a bin are given by $I_{ai}$ and $a_i$ respectively, and all bins have unit thickness. In a rainfall event, only the bins with $I_{ai} \leq P$ are fully filled and contribute to runoff, whereas the


others are partially filled and do not contribute to runoff. The total amount of filled storage in $I_a$ also increases with $P$ until it reaches a constant value when the $I_{ai}$s of all land use types are fully filled and the whole watershed is contributing to the runoff.

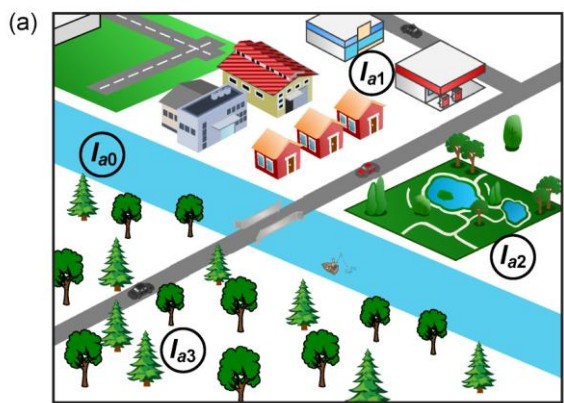

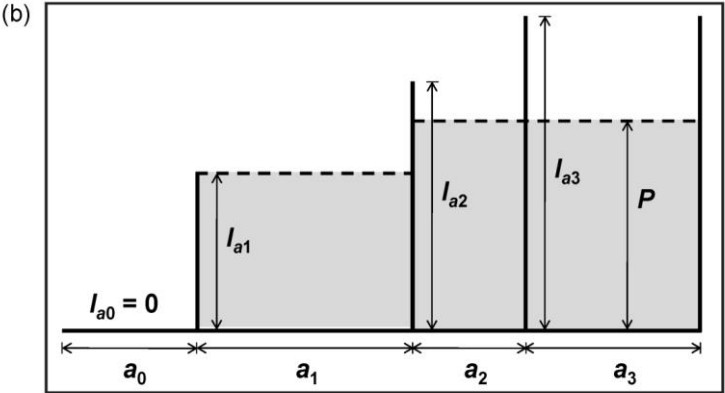

**Figure 3.** Spatial distribution of $I_a$ in a heterogeneous watershed (a) $I_{ai}$s of various HRUs mainly characterized by their land use types ($I_{a0} = 0 < I_{a1} < I_{a2} < I_{a3}$) (b) conceptual model in which each HRU is represented by a bin with height = $I_{ai}$, width = $a_i$, and unit thickness; shaded area indicates the filled portion during an event.

Consider a general case of a heterogeneous watershed with $n + 1$ HRUs such that

$$I_{a0} = 0 < I_{a1} < I_{a2} < .... < I_{an} \tag{12}$$

where $I_{a0}$ represents open water bodies and other surfaces that cannot abstract rainfall. The areal average of the total initial abstraction ($I_{aT}$) is given by





$$I_{aT} = \sum_{i=0}^{n} a_i \, I_{ai} \tag{13}$$

In a rainfall event, all the HRUs with $I_{ai} \leq P$ have their initial abstractions completely filled while the others are partially filled. Just by analyzing the runoff for that event, it is impossible to quantify the magnitudes of the $I_{ai}$s that are partially filled. Because they have not contributed to the runoff, all that can be said is that their $I_{ai}$s are greater than $P$ but their magnitudes remain

unknown. However, the information on the magnitudes of the $I_{ai}$s that are completely filled should be present in the runoff data. In other words, it takes larger rainfalls to fill larger $I_{ai}$s and gather information about their magnitude.

Then what is the effective initial abstraction of the watershed for a given rainfall event? Consider an event where the rainfall falls within the range: $I_{am} \leq P < I_{a(m+1)}$. HRUs with $I_{ai} \leq I_{am}$ have their initial abstractions completely filled and produce runoff, whereas HRUs with $I_{ai} \geq I_{a(m+1)}$ have their initial abstractions partially filled up to the level of $P$ and do not produce runoff.

The areal average of the filled portion (includes completely filled as well as partially filled HRUs) of the initial abstraction is given by

$$I_{aF} = \sum_{i=0}^{m} a_i \, I_{ai} + \left(1 - \sum_{i=0}^{m} a_i\right) P \tag{14}$$

The first term on the right-hand side of eq. (14) represents completely filled HRUs. The second term represents partially filled HRUs, all of which are filled to the level of $P$. Note that $I_{aT}$ is the areal average of total initial abstraction, whereas $I_{aF}$ is the

15 areal average of the filled portion. Therefore,

$$
\begin{aligned}
I_{aF} &< I_{aT} \quad \forall \quad P < I_{an} \\
I_{aF} &= I_{aT} \quad \forall \quad P \geq I_{an}
\end{aligned} \tag{15}
$$

The conceptual model presented in Figure 3, and in eqs. (14) and (15) is intuitively appealing, and also hints at the possibility that $I_{aW}$ may be equal to $I_{aF}$. This is because $I_{aF}$ increases with $P$ and approaches a constant value ($I_{aT}$), similar to the observations in Figure 2(c) and 2(d). Eq. (14) is also consistent with a distributed parameter model application of the CN

method as described in Section-3.

## 2.3. Variation of $I_{aF}$ with $P$

To investigate the variation of $I_{aF}$ with $P$, eqs. (14) and (15) are applied to the scenario presented in Figure 3, where $n = 3$. A plot of $I_{aF}$ vs. $P$ (Figure 4) shows that $I_{aF}$ increases with $P$ and becomes constant ($I_{aF} = I_{aT}$) at large values of $P$ ($P \geq I_{a3}$). The kink-points joining the line segments occur when the initial abstraction of one of the HRUs becomes completely filled. At

25 these points, $P$ is equal to one of the $I_{ai}$s. In between these points ($I_{am} < P < I_{a(m+1)}$), the relationship between $I_{aF}$ and $P$ is linear




with a slope of $\left(1 - \sum_{i=0}^{m} a_i\right)$. The slope abruptly changes across the kink-points. It decreases with $m$, and becomes zero when

$m = n$. The maximum value the slope can take is unity. This occurs with the line segment passing through the origin, when HRUs with zero initial abstraction are absent (i.e. $a_0 = 0$). When these HRUs are present, however, the origin itself is a kink-point where the slope abruptly jumps from unity to $1 - a_0$.

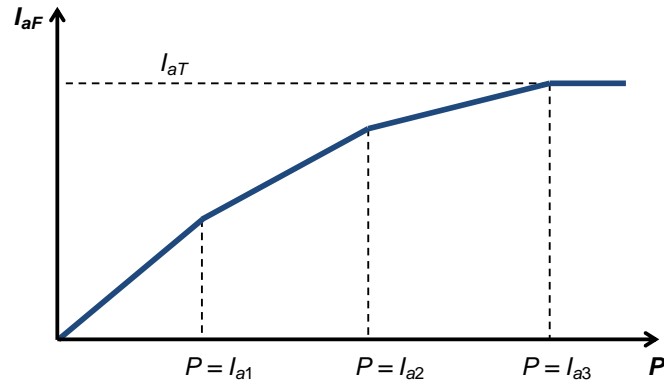

**Figure 4.** Variation of $I_{aF}$ [eqs.(14) and (15)] with $P$ for the scenario presented in Figure 3.

The analysis presented so far represents a discrete case where each HRU is homogeneous and has a finite area. The values of $I_{ai}$s vary discontinuously across the HRUs. Their areal distribution can be represented by a plot of $a_i$ vs. $I_a$ [Figure 5(a)]. The smaller the area of HRUs, the more numerous they are, and the more accurate is the representation of the heterogeneity within the watershed. The most ideal representation would occur when the HRUs shrink to points. Then the magnitudes of $I_{ai}$s within the watershed vary continuously and therefore can be represented by a probabilistic distribution of areal occurrence [Figure 5(b)]. It is impractical to characterize the watershed at such fine scale, but it is worth understanding the properties of the initial abstraction at the finest resolution first, and then making assumptions or simplifications later to suit the practical needs.

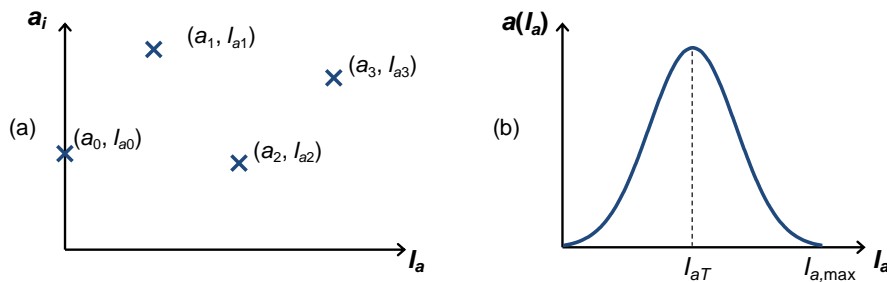

**Figure 5.** Representing areal distribution of $I_a$ within a watershed (a) discrete case (b) continuous case.




For the case of a continuous distribution of $I_a$, eq. (14) takes the form

$$I_{aF} = \int_0^P I_a \, a(I_a) \, dI_a + \left(1 - \int_0^P a(I_a) \, dI_a\right) P \tag{16}$$

where $a(I_a)$ is the probability density function of areal occurrence of $I_a$. The fractional area with initial abstraction $= I_a$ is given by $a(I_a) \, dI_a$. The upper limit of the integrals is set to $P$ because the last initial abstraction to completely fill up would be equal to $P$. The areal average of total initial abstraction, $I_{aT}$, is given by

$$I_{aT} = \int_0^{I_{a,\max}} I_a \, a(I_a) \, dI_a \tag{17}$$

where $I_{a,\max}$ is the maximum value of $I_a$ within the watershed. Thus, $I_{aT}$ is equal to the mean of the distribution [Figure 5(b)]. Eq. (15) then becomes

$$
\begin{aligned}
I_{aF} &< I_{aT} \quad \forall \quad P < I_{a,\max} \\
I_{aF} &= I_{aT} \quad \forall \quad P \geq I_{a,\max}
\end{aligned}
\tag{18}
$$

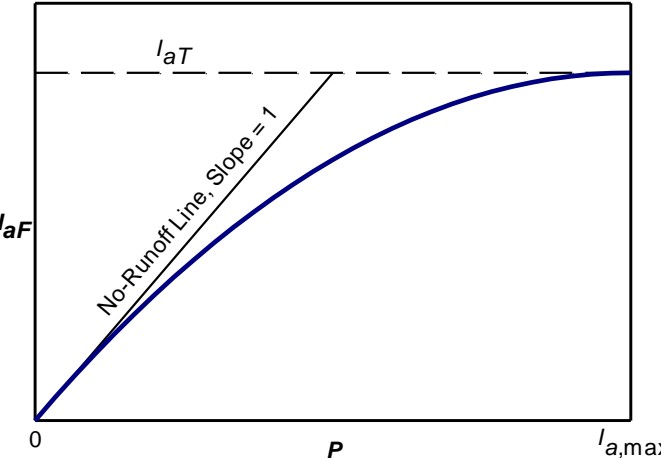

**Figure 6.** Variation of $I_{aF}$ with $P$ for a continuous distribution such as the one shown in Figure 5(b)

Unlike the discrete case, the slope of the $I_{aF}$ curve for the continuous case decreases smoothly with increasing $P$ (Figure 6). This is because the line segments in the discrete case (Figure 4) shrink to points in the continuous case. It follows from eq. (16) that




$$
\left.\frac{dI_{aF}}{dP}\right|_{P=0} = 1
$$

$$
\left.\frac{dI_{aF}}{dP}\right|_{P \geq I_{a,\max}} = 0
$$

(19)

Thus, the $I_{aF}$ curve is bounded by a line of slope = 1 passing through the origin, and a line of slope = 0 with the intercept equal to $I_{aT}$ (Figure 6). The line of slope = 1 is referred to as the no-runoff line because along this line $I_{aF} = P$. When the whole watershed is represented by a single HRU, the $I_{aF}$ curve coincides with the no-runoff line until $I_{aF} = I_{aT}$. A comparison of

5     Figure 6 to Figure 2(c) and 2(d) strengthens the case that $I_{aW}$ is equal to $I_{aF}$.

## 2.4. Variation of CN$_W$ with $P$

Let us hypothesize that $I_{aW} = I_{aF}$, i.e. the effective $I_a$ of a watershed is equal to the area-weighted average of the filled portion of the $I_{ai}$s. Then, if eq. (10) is written for CN$_W$, $I_a$ can be replaced by $I_{aF}$. Substituting eq. (16) in eq. (10) gives CN$_W$ as a function of $P$. When plotted against $P$, CN$_W$ starts at 100 when $P = 0$, and then decreases with increasing $P$ (Figure 7).

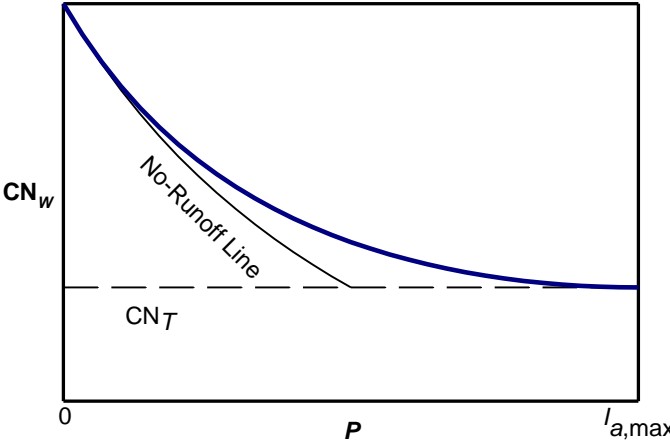

**Figure 7.** CN$_W$ as a function of $P$ when $I_{aW}$ is assumed to be equal to $I_{aF}$ (shown in Figure 6).

Differentiating eq. (10) and using eq. (19) gives

$$
\left.\frac{d(CN_W)}{dP}\right|_{P=0} = -\frac{10}{\lambda}
$$

$$
\left.\frac{d(CN_W)}{dP}\right|_{P \geq I_{a,\max}} = 0
$$

(20)



where the constant 10 has units of 1/in. Thus the $CN_W$ vs. $P$ curve is at its steepest at $P = 0$ and flattens with increasing $P$, and becomes constant when $P \geq I_{a,\max}$. This constant, $CN_T$, is the smallest value $CN_W$ can take and corresponds to the case $I_{aF} = I_{aT}$, when the initial abstractions of all the HRUs are fully filled. $CN_W$ as a function of $P$ is bounded by a curve corresponding to the condition $P = I_{aF}$, the no-runoff line, and a line of slope $= 0$ with the intercept equal to $CN_T$ (Figure 7).

The shape of the $CN_W$ vs. $P$ curve (Figure 7) generated using eqs. (10) and (16) is quite similar to the best-fit curves from field observations [Figure 2(a) and 2(b)]. Nearly 95% of the watersheds evaluated in the previous studies (D'Asaro and Grillone, 2012; Hawkins, 1993) also had responses identical to Figure 7, supporting the hypothesis that $I_{aW} = I_{aF}$. As pointed out by Soulis and Valiantzas (2012), *complacent behavior* appears to be a special case of *standard behavior* where observations from larger rainfalls are unavailable. Therefore, it is probably more appropriate to refer to any "CN decreasing with $P$" trend as
*standard behavior*, because it is caused by the inevitable presence of heterogeneity in a watershed.  It also shows that assuming a partial source area whenever a *complacent behavior* is observed (D'Asaro and Grillone, 2012; D'Asaro and Grillone, 2015) can be misleading.

### 2.5. $I_{aF}$ and $CN_W$ Curves for Various Distributions of $I_a$

The functional form of $a(I_a)$ defines the areal distribution of $I_a$ within a watershed. We considered idealized functional forms
of $a(I_a)$ that correspond to uniform, normal, triangular, and bi-modal distributions (Table 1). In each $a(I_a)$, the maximum or other key value was constrained so that the total area under the distribution was unity. For example, the y-coordinate of the apex in the triangular distribution must be equal to $2 / I_{a,\max}$ (Table 1). In the case of normal distribution, however, the area under the curve is unity only when the limits are infinite. Therefore, a standard deviation ($\sigma$) much less than $I_{a,\max}$ was used so that the area under the curve within the range $0 \leq I_a \leq I_{a,\max}$ is approximately equal to unity.

For each distribution, the corresponding functional form of $I_{aF}$ was determined using eq. (16) and the results are presented in Table 1. For the general case of $a(I_a)$ as a polynomial, the corresponding $I_{aF}$ is a polynomial two degrees higher than $a(I_a)$. For the normal distribution, $I_{aF}$ is a combination of Gaussian and Error functions (Table 1).

For the purpose of comparison, symmetrical versions of the distributions were considered such that all of them have the same minimum, mean, and maximum values of $I_a$ [Figure 8(a)]. The minimum value of $I_a$ was set to zero and the maximum value
was $I_{a,\max}$. Therefore, the mean for all the distributions was $I_{a,\max} / 2$.

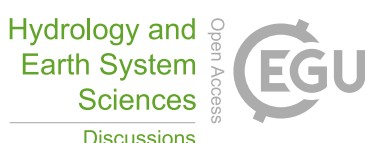



**Table 1.** Functional forms of $a(I_a)$ and $I_{aF}$ for various synthetic distributions

| Distribution | Graph | $a(I_a)$ | $I_{aF}$ |
|---|---|---|---|
| Uniform | | $\dfrac{1}{I_{a,\max}}$ | $P - \dfrac{P^2}{2I_{a,\max}}$ |
| Normal | | $\dfrac{1}{\sigma\sqrt{2\pi}}\, e^{-(I_a-\mu)^2/(2\sigma^2)}$ | $P - \dfrac{\sigma}{\sqrt{2\pi}}\left[ e^{-(P-\mu)^2/(2\sigma^2)} - e^{-\mu^2/(2\sigma^2)} \right] - \dfrac{(P-\mu)}{2}\left[ erf\left(\dfrac{P-\mu}{\sqrt{2}\sigma}\right) + erf\left(\dfrac{\mu}{\sqrt{2}\sigma}\right) \right]$ |
| Triangular | | $\dfrac{2I_a}{aI_{a,\max}}$ if $I_a \le a$ <br> $\dfrac{2}{(I_{a,\max}-a)}\left(1-\dfrac{I_a}{I_{a,\max}}\right)$ if $a < I_a$ | $P - \dfrac{P^3}{3aI_{a,\max}}$ if $P \le a$ <br> $\dfrac{1}{(I_{a,\max}-a)}\left[-\dfrac{a^2}{3} + I_{a,\max}P - P^2 + \dfrac{P^3}{3I_{a,\max}}\right]$ if $a < P$ |
| Bimodal | | $u$ if $I_a \le a$ <br> $0$ if $a < I_a < b$ <br> $v$ if $b \le I_a$ | $P - \dfrac{uP^2}{2}$ if $P \le a$ <br> $\dfrac{ua^2}{2} + (1-ua)P$ if $a < P < b$ <br> $\dfrac{(ua^2 - vb^2)}{2} + (1-ua+vb)P - \dfrac{vP^2}{2}$ if $b \le P$ |

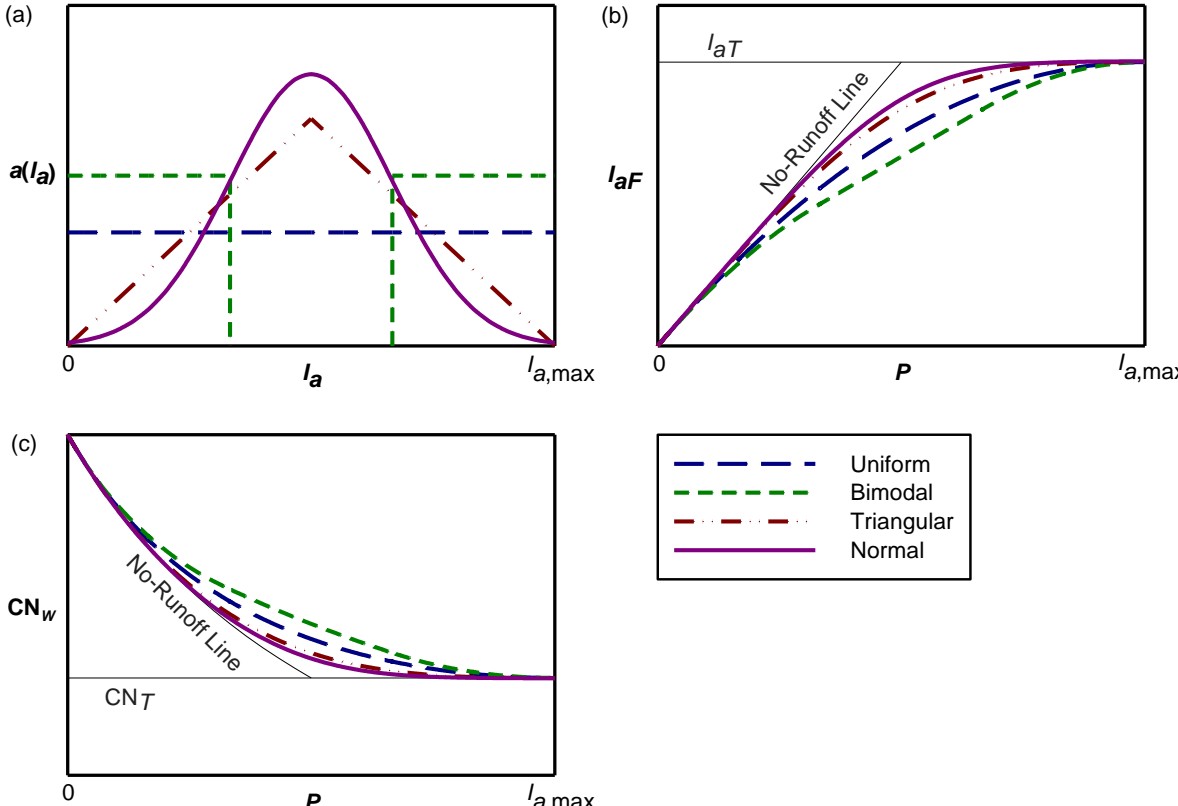

**Figure 8.** (a) Various symmetrical distributions of $I_a$ with the same minimum (zero), mean ($I_{a,max}/2$), and maximum ($I_{a,max}$), (b) the corresponding $I_{aF}$ curves calculated using eq. (16), (c) the corresponding $CN_W$ curves calculated using eq. (10).

The kurtosis (peakedness) of $a(I_a)$ has a major influence on the shapes of $I_{aF}$ and $CN_W$ plotted as functions of $P$ (Figure 8). The normal distribution has the greatest kurtosis whereas the bimodal distribution has the least. As the kurtosis decreases, the $I_{aF}$ and $CN_W$ curves deviate further from the bounding lines (Figure 8). When there is a gap in the distribution, as in the case of the bimodal distribution, the corresponding $I_{aF}$ curve is linear for the range spanning the gap. This is consistent with the discrete case where $I_{aF}$ was represented by line segments for the gaps in between the discrete values of $I_{ai}$ (Figure 4).

Skewness of $a(I_a)$ also affects $I_{aF}$, and this is illustrated by an idealized case where an initially uniform distribution is positively skewed [Figure 9(a) and 9(b)]. The mean of $a(I_a)$, which is equal to $I_{aT}$ [eq. (17)], decreases with increasing positive skewness. This is important because a land use change such as conversion of forest to urban land is expected to increase the positive skewness (i.e. more low values of $I_a$). During the conversion, $I_{a,max}$ remains unchanged while some forested land remains. When the entire forest is converted, $I_{a,max}$ drops to a lower value.



The analysis also shows that a watershed cannot be characterized or compared with other watersheds using a single value of CN [such as $CN_\infty$ used in asymptotic fitting, eq. (7)]. Depending on the distribution of heterogeneity, the relative runoff potential of a watershed can be $P$ dependent. This is illustrated by considering two uniform distributions, uni1 and uni2, where uni2 has a narrower range and a smaller mean than uni1 [Figure 9(c)]. For smaller values of $P$, $I_{aF,uni1} < I_{aF,uni2}$ [Figure 9(d)], and therefore $CN_{W,uni1} > CN_{W,uni2}$. However, for larger values of $P$, the converse is true. Thus, the watershed with uni1 generates more runoff for smaller values of $P$, whereas the watershed with uni2 generates more runoff for larger values of $P$.

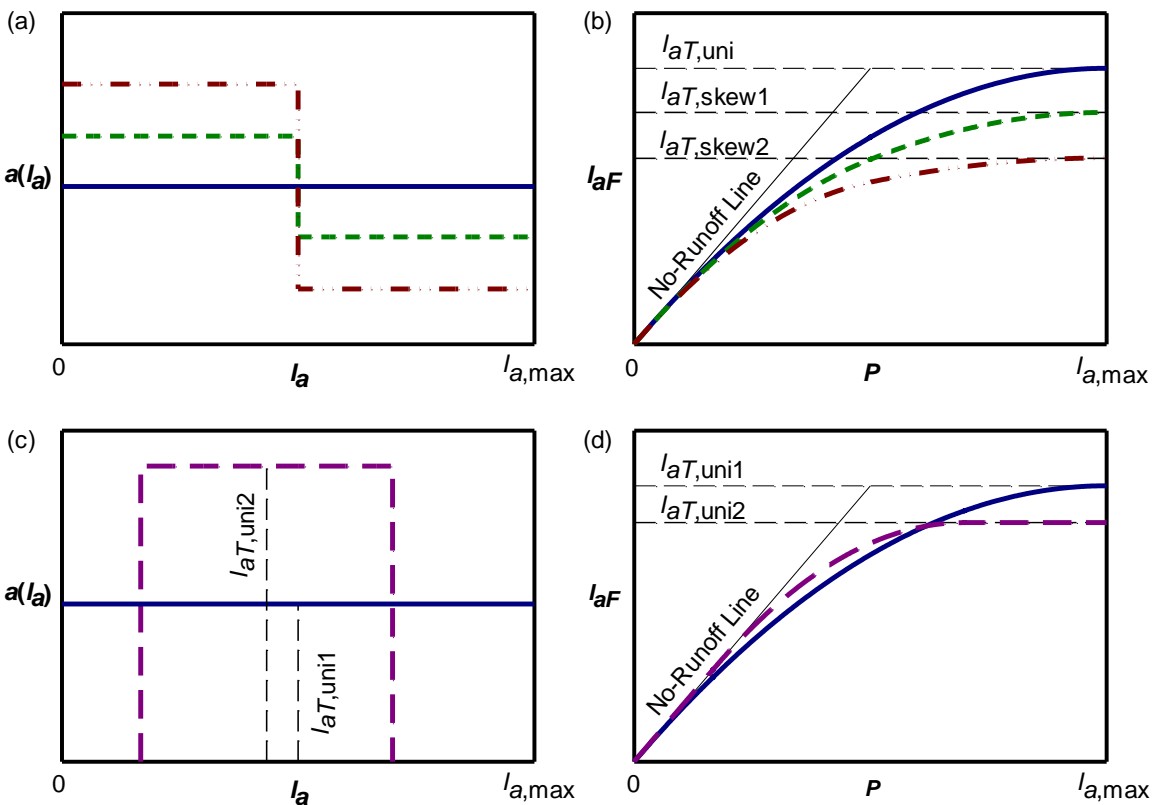

**Figure 9.** Effect of skewness, mean, and range of $a(I_a)$ on $I_{aF}$ (a) uniform, uni (solid), and two positively skewed distributions, skew1 (dashed) and skew2 (dash dot dot) (b) $I_{aF}$ as a function of $P$ for the distributions shown in 9a (c) uniform distributions uni1 (solid) and uni2 (dashed) where uni2 has a narrower range of values of $I_a$ and a smaller mean than uni1 (d) $I_{aF}$ as a function of $P$ for the distributions shown in 9c.




## 3. Effect of Heterogeneity on $S$

Similar to the case of $I_a$, the presence of heterogeneity also causes the effective $S$ of a watershed ($S_W$) to vary with $P$. Functional form of $S_W$ depends not only on the potential maximum retentions of the HRUs ($S_i$s) but also on the $I_{ai}$s. $S_W$ can be estimated using eq. (2) if the quantities $I_{aW}$, $Q_W$, and $F_W$ are known. A distributed modeling approach can be used to calculate these quantities for a heterogeneous watershed. Distributed CN models, e.g. SWAT (Gassman et al., 2007), commonly calculate $Q_W$ as the area-weighted average of $Q_i$s, and this assumption can also be extended to $F_W$. Thus,

$$Q_W = \sum_{i=0}^{n} a_i \, Q_i$$

$$F_W = \sum_{i=0}^{n} a_i \, F_i$$

(21)

Using eq. (21) and applying mass balance [eq. (1)] at watershed and HRU scales gives eq. (14) for $I_{aW}$. This shows that $I_{aW}$ calculated using a distributed model is equal to $I_{aF}$.

Writing an expression for $S_W$ in terms of $I_{ai}$s and $S_i$s for a general case of a heterogeneous watershed is cumbersome. Therefore, it is only presented graphically for an example of a heterogeneous watershed. However, an expression for $S_W$ can be presented in a compact form for a special case where all the $I_{ai}$s are zero as

$$\text{if} \quad I_{ai} = 0 \quad \forall \quad i \in \{0,1,....n\}$$

(22)

$$S_W = \frac{\displaystyle\sum_{i=0}^{n} \frac{a_i S_i}{P + S_i}}{\displaystyle\sum_{i=0}^{n} \frac{a_i}{P + S_i}}$$

Thus, $S_W$ varies from the area-weighted harmonic mean $\left(\displaystyle\sum_{i=0}^{n} \frac{a_i}{S_i}\right)^{-1}$ when $P = 0$, to the area-weighted arithmetic mean $\left(\displaystyle\sum_{i=0}^{n} a_i S_i\right)$ when $P \gg S$.

To illustrate the effect of heterogeneity on $S_W$, an example watershed with the storage distribution shown in (Table 2) was considered. The variation of $S_W$ with $P$ was analyzed for the cases of $\lambda_i = 0$ and $\lambda_i = 0.2$ (Figure 10). In both cases, $S_W$ increases with $P$ and approaches the area-weighted arithmetic mean, $S_\infty$, for large values of $P$. In the case of $\lambda_i = 0$, the slope of the curve





is maximum at the origin, and decreases monotonically with $P$. In case of $\lambda_i = 0.2$, however, the slope is zero at the origin and generally increases with $P$ until $P \approx I_{an} = 40$ mm ($P \approx I_{a,\max}$ for the continuous case), where it reaches its maximum value. Thereafter the slope decreases monotonically with $P$, giving an S-shaped curve. In other words, the slope generally increases with $P$ until the entire watershed area contributes to the runoff, and decreases thereafter.

5 **Table 2.** Storage distribution in a hypothetical heterogeneous watershed used to illustrate the variation of $S_W$ with $P$.

| HRU | $a_i$ | $S_i$ (mm) |
|-----|-------|------------|
| 0 | 0.05 | 0 |
| 1 | 0.20 | 50 |
| 2 | 0.35 | 100 |
| 3 | 0.25 | 150 |
| 4 | 0.15 | 200 |

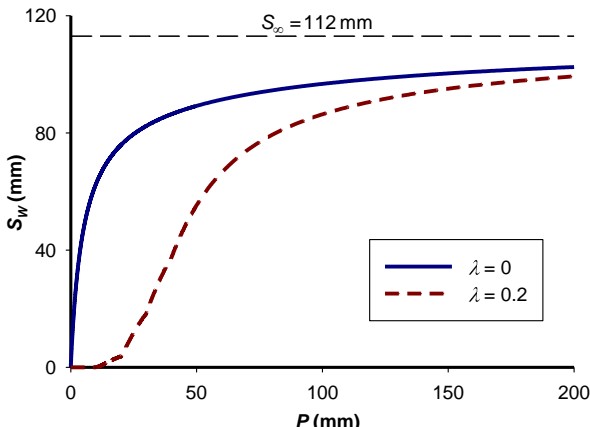

**Figure 10.** Variation of $S_W$ with $P$ in a heterogeneous watershed with the storage distribution shown in Table 2.

The similarities between $I_{aW}$ and $S_W$ are that they both increase with $P$ and have an upper limit equal to the area-weighted
10  arithmetic mean of their respective components. The difference is that $I_{aW}$ reaches its upper limit of $I_{aT}$ for a finite value of $P$ ($P = I_{an}$ or $P = I_{a,\max}$), whereas $S_W$ requires large values of $P$ ($P \gg S$) to reach its upper limit of $S_\infty$. Moreover, $S_W$ vs. $P$ is an S-





shaped curve when $\lambda_i > 0$. This shows that $I_{aW}$ and $S_W$ are not proportional, i.e. $\lambda_W$ is not a constant even though $\lambda_i$s are assumed to be equal and constant.

## 4. Application

The analysis from previous sections shows that $I_{aW}$ and $S_W$ are functions of $P$, and gives their functional forms. Incorporating these functions in the lumped parameter application can potentially improve the performance of the CN method.

### 4.1. $I_{aW}$ as a function of $P$

The distributed parameter modeling approach, eq. (21) with the application of mass balance [eq. (1)] at watershed and HRU scales, shows that $I_{aW} = I_{aF}$. $I_{aF}$ is given by eq. (14) for the discrete case and eq. (16) for the continuous case. All the distributions in Table 1, except the normal distribution, gave a zero-intercept polynomial for $I_{aF}$. Therefore, using a quadratic function of the form

$$
\begin{aligned}
I_{aW} &= c_1 P - c_2 P^2 & \forall \quad P \leq I_{a,\max} \\
I_{aW} &= I_{aT} = c_1(I_{a,\max}) - c_2(I_{a,\max})^2 & \forall \quad P > I_{a,\max}
\end{aligned}
\tag{23}
$$

is an efficient way to describe $I_{aW}$. In eq. (23), $c_1$ and $c_2$ are calibration parameters such that $0 \leq c_1 \leq 1$ and $c_2 \geq 0$. Since the slope of $I_{aW}$ is zero at $P = I_{a,\max}$ [eq. (19)], it follows from eq. (23) that

$$
\begin{aligned}
I_{a,\max} &= \frac{c_1}{2c_2} \\
I_{aT} &= \frac{c_1^2}{4c_2}
\end{aligned}
\tag{24}
$$

Similarly, the slope of $I_{aW}$ is unity at $P = 0$ so $c_1$ should be unity. However, it was kept as a free parameter in eq. (23) to allow for the approximation of piecewise functions (e.g. $I_{aF}$ for triangular and bimodal distributions in Table 1). Moreover, the analysis for the discrete case shows that when HRUs with zero initial abstraction are present, the origin is a kink-point where the slope abruptly jumps from unity to $1-a_0$. To avoid over-parameterization of the model, a polynomial of degree > 2 for $I_{aW}$ was not considered.

### 4.2. $S_W$ as a function of $P$

The sigmoid shaped function of $S_W$, with the conditions that $S_W = 0$ when $P = 0$ and that the maximum slope occurs at $P = I_{a,\max}$, requires at least two parameters to describe it. However, this along with eq. (23) would also increase the number of calibrated parameters in the CN method, increasing its complexity and potentially causing non-uniqueness. A relatively simple approach




is to assume that $S_W$ is constant similar to the conventional CN method. Another approach is to assume that $S_W$ is proportional to $I_{aW}$, i.e. eq. (3) is applicable for a heterogeneous watershed.

Here the emphasis is placed on treating $I_{aW}$ as a function of $P$ while offering some flexibility on how $S_W$ is treated. This is because the variation of $I_{aW}$ with $P$ had a significant impact on the model performance, whereas including the variation of $S_W$

with $P$ showed only marginal or no improvement. This may be because $I_{aW}$ is a component of mass balance [eq. (1)] while $S_W$ is not. $F_W$, which is the filled portion of $S_W$, is a component of mass balance and varies with $P$ even if $S_W$ is assumed to be a constant. Therefore, to maintain the simplicity of the CN method and avoid the problems of over-parameterization and non-uniqueness, modeling the sigmoid-shaped function of $S_W$ is omitted.

### 4.3. Lumped Parameter Models

Lumped parameter application of the CN method was modified by treating $I_{aW}$ as a function of $P$ as described in the previous section. Modified lumped parameter CN models were evaluated by comparing their performance with that of the conventional lumped parameter CN models.

#### *4.3.1. Conventional Models (CMs)*

Conventional CN models are defined by eqs. (1) through (5), and by the assumption that $I_{aW}$ and $S_W$ are independent of $P$. In

this study two types of conventional models, referred to as CM0.2 and CMλ, were used. In CM0.2, $\lambda_W$ was fixed at 0.2, and in CMλ, $\lambda_W$ was determined by calibration. Thus CM0.2 had one free parameter, $S_W$, whereas CMλ had two free parameters, $\lambda_W$ and $S_W$.

#### *4.3.2. Variable Initial Abstraction Models (VIMs)*

VIMs are defined by eqs. (1), (2), (4), (5), and (23), and they have three free parameters. If $S_W$ is assumed to be independent

of $P$, then the model requires calibration of $c_1$, $c_2$, and $S_W$, and is referred to as VIMS. If eq. (3) is also included, then the model requires calibration of $c_1$, $c_2$, and $\lambda_W$, and is referred to as VIMλ.

### 5. Evaluation

Lumped parameter models described in the previous section were evaluated in their ability to predict runoff and account for watershed heterogeneity. Accounting for heterogeneity means that the model accurately predicts $I_{aW}$ and $S_W$, and runoff from

smaller events. This is because (i) $I_{aW}$ and $S_W$ as functions of $P$ are directly related to heterogeneity, and (ii) inability to account for their variation with $P$ causes under-estimation of runoff in smaller events.





Evaluation of lumped parameter models requires the data for $I_{aW}$, $Q_W$ and $S_W$. This is generated using a distributed parameter model application of the CN method. The assumption is that a distributed parameter model accounts for heterogeneity, and therefore its estimates of $I_{aW}$, $Q_W$ and $S_W$ are accurate.

## 5.1. Distributed Parameter Model

In a distributed parameter model, eqs. (1) through (5) are applicable at the HRU scale, with the assumption that $I_{ai}$ and $S_i$ are independent of $P$. Once $Q_i$ and $F_i$ are calculated for each HRU, watershed scale quantities $I_{aW}$, $Q_W$, $F_W$ and $S_W$ are calculated using eqs. (14), (21), and (2).

The distributed parameter model was applied to an idealized synthetic watershed with the storage distribution shown in Table 2, for the cases of $\lambda_i = 0$, 0.2, and 0.5. A range of values of $P$ were synthetically generated such that they vary lognormally from 0.1 mm to 200 mm with a median of 8 mm. For each rainfall event, $I_{aW}$, $Q_W$, $F_W$ and $S_W$ were calculated, and used in the evaluation of the lumped parameter models.

The reason for using a synthetic watershed here is that the heterogeneity can be precisely defined and used to evaluate the predictions of heterogeneity by the lumped parameter models. In real watersheds the heterogeneity has to be determined by calibration, and there can be non-uniqueness when multiple HRUs are present. Application of these modified models to data from real watersheds is discussed by Santikari and Murdoch (2018).

## 5.2. Model Evaluation Criteria

Each lumped parameter model was calibrated by minimizing the sum of the squared residuals between its predicted runoff ($Q_W$) and the baseline from the distributed parameter model. All the models were evaluated using the Nash-Sutcliffe efficiency parameter (NSE), the standard error of estimate (SEE), and the percent bias (PB) (McCuen, 2003; Moriasi et al., 2007). NSE can vary from $-\infty$ to 1. The calculations and observations are exactly equal when NSE = 1. The calculations are only as good as the average observation when NSE = 0. SEE is the root-mean-square residual adjusted to the degrees of freedom (Santikari and Murdoch, 2018). A smaller SEE indicates a better performance, and its ideal value is zero. PB indicates whether the model is over (PB < 0) or under-predicting (PB > 0) on average. The optimal value for PB is zero.

NSE values were calculated for the model predictions of runoff ($NSE_Q$), initial abstraction ($NSE_{Ia}$), potential maximum retention ($NSE_S$), and runoff from events with $P$ less than the median value ($NSE_{Q50}$). PB values were calculated for runoff from all the events ($PB_Q$) and runoff from events with $P$ less than the median value ($PB_{Q50}$). SEE was calculated for runoff from all the events ($SEE_Q$).

$NSE_{Ia}$ and $NSE_S$ indicate how accurately a lumped parameter model predicts the watershed heterogeneity. $NSE_Q$, $SEE_Q$, and $PB_Q$ reflect the overall accuracy in a model prediction of runoff from all the events, whereas $NSE_{Q50}$ and $PB_{Q50}$ reflect the





accuracy in predicting runoff from smaller events ($P < 8$ mm). Conventional models tend to under-predict runoffs from smaller events because of the usage of constant $I_a$ and $S$. They often falsely predict zero-runoffs because the runoff condition ($P > I_a$) cannot be overcome in smaller events. $NSE_{Q50}$ and $PB_{Q50}$ are used to expose this shortcoming.

## 6. Results and Discussion

The results show that using variable initial abstraction improved the accuracy of model predictions of runoff and heterogeneity (Table 3). Based on their overall performance, the models can be arranged from the best to the worst as VIMλ > VIMS > CMλ > CM0.2. Results for the case of $\lambda_i = 0$ are not presented in Table 3 because VIMλ, VIMS, and CMλ performed equally well while CM0.2 was the worst (i.e. VIMλ = VIMS = CMλ > CM0.2).

Variable $I_a$ models predicted runoff better than the conventional models. It was not possible to determine relative model performance using $NSE_Q$ because it was 1.0 for all the models. This was because $NSE_Q$ was strongly influenced by a few larger events. A good fit in these events was sufficient to render $NSE_Q = 1.0$, and therefore it is not listed in Table 3. However, $SEE_Q$ decreased down the table, indicating an improvement in performance. $PB_Q$ was positive for all the models, indicating that they all under-predicted runoff. The extent of under-prediction, however, was smaller in variable $I_a$ models than the conventional models.

Variable $I_a$ models gave a better estimate of watershed heterogeneity than the conventional models as indicated by the higher values of $NSE_{Ia}$ and $NSE_S$ (Table 3). $NSE_{Ia}$ was zero or negative in the conventional models, whereas it varied from 0.2 to 0.7 in the variable $I_a$ models. $NSE_S$ was negative in all the models, indicating that their estimates of $S$ were poor. In case of the conventional models this was due to using uniform $I_a$ and $S$, and thereby homogenizing the watershed. In case of the variable $I_a$ models, this was due to their inability to model the S-shaped function of $S$. Based on $NSE_{Ia}$ and $NSE_S$, VIMλ was the best model in estimating watershed heterogeneity.

Variable $I_a$ models also predicted runoff better than the conventional models in smaller rainfall events ($P < 8$ mm) as indicated by $NSE_{Q50}$ and $PB_{Q50}$. In both cases of $\lambda_i = 0.2$ and 0.5, only HRU #0 (Table 2) produced runoff when $P < 8$ mm. This was similar to the case of a partial source area. As CM0.2 and CMλ predicted an $I_a > 8$ mm in both the cases (Table 3), they falsely predicted zero-runoffs in all the events with $P < 8$ mm because the runoff condition ($P > I_a$) could not be overcome. Therefore, their $PB_{Q50} = 100$ in both the cases, indicating a 100% under-prediction in small events. Their $NSE_{Q50}$ was also poor with the same value in both the cases. VIMS performed slightly better than the conventional models with 70-90% under-predictions, and with $NSE_{Q50}$ varying from -0.8 to -1.8 (Table 3). VIMλ performed significantly better than all the other models with 30% or less under-predictions, and with $NSE_{Q50}$ varying from 0.6 to 0.9. Even though there were under-predictions, there was no false prediction of zero-runoff for any of the events in the variable $I_a$ models.



**Table 3.** The performance of lumped parameter CN models that were calibrated to the runoff data generated using a distributed CN model for two cases of a synthetic watershed with the storage distribution shown in Table 2. SEE, $I_a$, and $S$ are in mm. (SEE: Standard Error of Estimate, PB: Percent Bias, NSE: Nash-Sutcliffe Efficiency parameter)

| Lumped Model | $SEE_Q$ | $PB_Q$ | $NSE_{Ia}$ | $NSE_S$ | $NSE_{Q50}$ | $PB_{Q50}$ | $\lambda_W$ | $I_a$ or $I_{aT}$ | $I_{a,\max}$ | $S$ or $S_\infty$ |
|---|---|---|---|---|---|---|---|---|---|---|
| Distributed Model: $\lambda_i = 0.2$, $I_{aT} = 22$, $I_{a,\max} = 40$, $S_\infty = 112$ | | | | | | | | | | |
| CM0.2 | 0.91 | 12.6 | -1.8 | -13 | -2.9 | 100 | 0.20 | 19 | - | 97 |
| CMλ | 0.37 | 5.4 | 0.0 | -26 | -2.9 | 100 | 0.07 | 9 | - | 132 |
| VIMS | 0.13 | 2.1 | 0.2 | -22 | -0.8 | 71 | - | 12 | 64 | 121 |
| VIMλ | 0.06 | 0.2 | 0.4 | -3 | 0.9 | 16 | 0.09 | 11 | 43 | 124 |
| Distributed Model: $\lambda_i = 0.5$, $I_{aT} = 56$, $I_{a,\max} = 100$, $S_\infty = 112$ | | | | | | | | | | |
| CM0.2 | 0.81 | 18.8 | -1.4 | -102 | -2.9 | 100 | 0.20 | 31 | - | 155 |
| CMλ | 0.66 | 13.6 | -0.3 | -166 | -2.9 | 100 | 0.11 | 21 | - | 197 |
| VIMS | 0.26 | 6.9 | 0.7 | -83 | -1.8 | 87 | - | 37 | 130 | 140 |
| VIMλ | 0.13 | 1.6 | 0.7 | -9 | 0.6 | 33 | 0.21 | 33 | 96 | 153 |





In the models where $\lambda_W$ was calibrated (CMλ and VIMλ), it was smaller than $\lambda_i$ (Table 3). This shows that $\lambda$ at the watershed scale tends to be smaller than that at the HRU scale in the lumped parameter models. All the models under-predicted $I_a$ or $I_{aT}$ with CMλ being the most severe. There was also a corresponding over-prediction of $S$ or $S_\infty$ by all the models except for the case of $\lambda_i = 0.2$ in CM0.2. Again, the most over-prediction of $S$ occurred in CMλ. The under-prediction of $I_a$ and the corresponding over-prediction of $S$ is due to the transfer of storage from $I_a$ to $S$, which generally improves the performance in the conventional models.

### 6.1. Storage Transfer from $I_a$ to $S$

The storage in a watershed is distributed between $I_a$ and $S$. $I_a$ is the part of the storage that does not produce runoff while being filled, whereas $S$ is the part that produces runoff while being filled. Using eqs. (2) and (1) it can be shown that

$$S = \frac{(P - I_a)(P - I_a - Q)}{Q} \tag{25}$$

For an observed storm event, $P$ and $Q$ are known and therefore are constants in eq. (25), so decreasing $I_a$ will increase $S$. However, the magnitude of increase in $S$ will be greater than the magnitude of decrease in $I_a$. This is illustrated by differentiating eq. (25) and using eq. (4) to give

$$\frac{dS}{dI_a} = -\left(1 + \frac{2S}{P - I_a}\right) \quad \forall \quad P > I_a \tag{26}$$

Thus, $dS/dI_a$ is always negative and less than or equal to -1. If $(P - I_a) \gg S$ or $S \approx 0$, then $dS/dI_a \approx -1$, implying an equal transfer in storage between $I_a$ and $S$. However, as $P$ decreases, $dS/dI_a$ becomes less than -1, implying that $S$ changes more rapidly than $I_a$. In other words, the relative change of magnitude in $S$ with respect to $I_a$ is large for smaller $P$, decreases with increasing $P$, and approaches unity for large values of $P$.

Storage transfer is evident when the values of $I_a$ and $S$ for the models CM0.2 and CMλ are compared (Table 3). For the case of $\lambda_i = 0.2$, $I_a$ decreased from 19 mm in CM0.2 to 9 mm in CMλ, whereas $S$ increased from 97 mm to 132 mm, i.e. $dS/dI_a = -3.5$. Similarly for the case of $\lambda_i = 0.5$, $dS/dI_a = -4.2$.

A transfer of storage from $I_a$ to $S$ improves the performance in the conventional models (i.e. CMλ > CM0.2) because (i) a smaller $I_a$ reduces the percentage of events with falsely predicted zero-runoffs, and (ii) it allows the model to mimic a variable $I_a$. Because of a larger $I_a$, CM0.2 falsely predicted zero-runoffs in 80% of the events for $\lambda_i = 0.2$, and in 85% of the events for $\lambda_i = 0.5$. In case of CMλ they dropped to 57% and 81% respectively because its $I_a$ was smaller than CM0.2. Mimicking variable $I_a$ can be explained by considering $I_{aF}$ and $F$, which are the filled portions of $I_a$ and $S$ respectively. $I_{aF}$ and $F$ have similar



functional relationships with $P$ (compare Figure 6 to Figure 1), i.e. they both increase with $P$ and approach a constant for large values of $P$. In the conventional CN models, there is no provision to represent $I_{aF}$ as a function of $P$. However, $F$ is understood to be a function of $P$ and is treated as such through eq. (2) and Figure 1. Therefore, by transferring the storage from $I_a$ to $S$, CMλ uses $F$ as a surrogate for $I_{aF}$, thereby partly mimicking the variable nature of $I_{aF}$.

Storage transfer from $I_a$ to $S$ also causes a decrease in $\lambda_W$ (Table 3). Conversely, when $\lambda_W$ decreases, storage is transferred from $I_a$ to $S$. This is important because several studies (Baltas et al., 2007; D'Asaro and Grillone, 2012; Shi et al., 2009; Woodward et al., 2003) found that the optimal value of $\lambda_W$ was much less than 0.2, and even close to zero in many watersheds. This shows that there is a positive correlation between a decrease in $\lambda_W$, storage transfer from $I_a$ to $S$, and a general increase in model performance for the reasons mentioned above.

**6.2. Model Suitability**

One of the main objectives of this study was to improve the predictive ability of the CN method while maintaining its simplicity. Using the number of calibrated parameters as an indicator, the models can be arranged in the order of increasing complexity as: CM0.2 (one) < CMλ (two) < VIMS = VIMλ (three). CM0.2 was the simplest, but also had the poorest performance (Table 3). Moreover, there is no justification in fixing $\lambda_W$ at 0.2 or any other constant as its optimal value can vary
from zero to one (Hawkins et al., 2008). Therefore, the usage of CM0.2 is not recommended.

CMλ predicted the overall runoff and the runoff from small events better than CM0.2. Often, the optimal $\lambda_W$ is much smaller than 0.2 and this allows CMλ to partly mimic a variable $I_{aF}$ by transferring storage from $I_a$ to $S$. A smaller $\lambda_W$ also reduces the false prediction of zero-runoffs, which are completely eliminated when $\lambda_W = 0$. Compared to the variable $I_a$ models, CMλ is a poor predictor of runoff and watershed heterogeneity (Table 3). However, in watersheds with negligible $I_{ai}$s (or $\lambda_i \approx 0$) CMλ
can perform as well as the variable $I_a$ models, and therefore may be preferable because of its simplicity.

Variable $I_a$ models show that significant improvement in the model prediction of overall runoff and heterogeneity can be achieved by using one extra parameter (Table 3). This is because the functional form of $I_{aF}$ [eq. (23)] is consistent with the observations [Figure 2(c) and 2(d)] and the results from the theoretical analysis of heterogeneous watersheds [eq. (16), Figure 6, and Table 1]. Using variable $I_a$ also improved the runoff predictions in small events and eliminated the false prediction of
25 zero-runoffs. Therefore, their application is recommended in heterogeneous watersheds with non-zero initial abstractions.

**7. Conclusions**

Watershed heterogeneity causes calculated values of $I_a$, $S$, and CN to vary with $P$. Therefore, using a single effective value of these quantities at the watershed scale can lead to systematic errors in the predictions of $Q$. This problem can be mitigated by treating $I_a$, $S$, or CN as functions of $P$. A theoretical analysis assuming spatial variation of $I_a$ led to the following conclusions.





1. *Effective $I_a$ of a watershed is equal to the filled portion of the total storage in $I_a$*

The total storage (called $I_{aT}$) is constant, whereas the filled portion (called $I_{aF}$) is a function of $P$ [eq. (16)]. Variation of $I_{aF}$ with $P$ (Figure 6) is similar to the variation of calculated $I_a$ (also called effective $I_a$ or $I_{aW}$) with $P$ [Figure 2(c) and 2(d)]. This shows that $I_{aW} = I_{aF}$, which is also supported by a distributed model using many HRUs [eq. (21)]. The form of $I_{aF}$ as a function of $P$ depends on the spatial distribution of $I_a$ within a watershed (Table 1, Figures 8 and 9).

2. *$\lambda$ decreases with increasing spatial scale*

Using the definition of $I_a$ and $CN_W$, calculated as the area-weighted average of $CN_i$s (CNs of the HRUs), it can be shown that $\lambda_W < \lambda_i$ [eqs. (8) through (11)]. Even when $\lambda_W$ was calibrated using $CM\lambda$, the result was $\lambda_W < \lambda_i$ (Table 3). This shows that in conventional models, $\lambda$ at the watershed scale tends to be smaller than that at the HRU scale, i.e. $\lambda$ decreases with increasing spatial scale.

3. *Replacing $I_a$ with $I_{aF}$ can account for heterogeneity*

Heterogeneity causes the effective $I_a$ of a watershed to vary with $P$, so to account for heterogeneity variable $I_a$ models (VIMs) replace $I_a$ with $I_{aF}$, which is a function of $P$ (Figure 6). For practical purposes, $I_{aF}$ can be treated as a quadratic function of $P$ [eq. (23)] with two free parameters $c_1$ and $c_2$ that need to be calibrated. In addition, the model also requires the calibration of either $S$ (VIMS) or $\lambda$ (VIM$\lambda$).

4. *Variable $I_a$ models perform better than the conventional models*

Variable $I_a$ models predict runoff and heterogeneity better than the conventional models CM0.2 *($\lambda = 0.2$)* and CM$\lambda$ (calibrated $\lambda$). They also eliminate the false prediction of zero-runoffs and improve runoff predictions in small events. Based on their overall performance, the models are arranged from the best to the worst as VIM$\lambda$ > VIMS > CM$\lambda$ > CM0.2.

5. *Storage transfer can improve model performance*

Storage transfer from $I_a$ to $S$ generally improves the model performance because the filled portions of $I_a$ and $S$, $I_{aF}$ and $F$ respectively, have similar functional relationships with $P$ (compare Figure 6 to Figure 1). This enables a CN model to partly mimic a variable $I_{aF}$ by using $F$ as its surrogate. Storage transfer also lowers the threshold $P$ for runoff generation, thereby reducing the false prediction of zero-runoffs. Storage transfer decreases $\lambda_W$ [eq. (3)], and this can explain why the optimal value of $\lambda_W$ from published studies is much less than 0.2 or even zero in many watersheds.




**Acknowledgments**

Primary funding for this study was provided by the USDA Natural Resources Conservation Service (NRCS-69-4639-1-0010) through the Changing Land Use and Environment (CLUE) Project at Clemson University. Additional support was provided by the USDA Cooperative State Research, Education, and Extension Service under project number SC-1700278.

5    **List of Symbols**

| | | |
|---|---|---|
| $a_i$ | = | fractional area of the $i^{th}$ HRU |
| $a(I_a)$ | = | probability density function of areal occurrence of $I_a$ |
| CM0.2 | = | conventional curve number model with $\lambda = 0.2$ |
| CM$\lambda$ | = | conventional curve number model with calibrated $\lambda$ |
| CN | = | curve number, applicable to any spatial scale |
| $CN_i$ | = | curve number of the $i^{th}$ HRU |
| $CN_T$ | = | curve number of a watershed when $I_{aF} = I_{aT}$ |
| $CN_W$ | = | curve number of a watershed |
| $F$ | = | cumulative infiltration after runoff begins |
| HRU | = | hydrologic response unit |
| $I_a$ | = | initial abstraction, applicable to any spatial scale |
| $I_{aF}$ | = | areal average of the filled portion of $I_{aT}$ |
| $I_{ai}$ | = | initial abstraction of the $i^{th}$ HRU |
| $I_{aT}$ | = | areal average of the total initial abstraction |
| $I_{aW}$ | = | effective initial abstraction of a watershed |
| $I_{a,max}$ | = | maximum value of $I_a$ within a watershed |
| $\lambda$ | = | initial abstraction ratio, applicable to any spatial scale |
| $\lambda_i$ | = | initial abstraction ratio at HRU scale |




| $\lambda_W$ | = | initial abstraction ratio at watershed scale |
|---|---|---|
| $m$ | = | no. of HRUs with fully filled non-zero $I_{ai}$s |
| $n$ | = | no. of HRUs with non-zero $I_{ai}$s |
| NSE | = | Nash-Sutcliffe efficiency parameter |
| $P$ | = | event rainfall |
| PB | = | percent bias |
| $Q$ | = | event runoff |
| $R^2$ | = | coefficient of determination |
| $S$ | = | potential maximum retention, applicable to any spatial scale |
| $S_i$ | = | potential maximum retention of $i^{th}$ HRU |
| $S_\infty$ | = | maximum value of $S_W$, occurs when $P$ is infinitely large |
| $S_W$ | = | effective potential maximum retention of a watershed |
| SEE | = | standard error of estimate |
| VIM$\lambda$ | = | variable initial abstraction model in which $\lambda$ is calibrated |
| VIMS | = | variable initial abstraction model in which $S$ is calibrated |

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
