# Peer review of "Including Effects of Watershed Heterogeneity in the Curve Number Method Using Variable Initial Abstraction"

_Hydrology and Earth System Sciences, 2017_

## Referee Comment (RC1) · Anonymous Referee #1 · 19 Dec 2017

GENERAL COMMENTS The authors present a methodology to account for heterogeneity in the calibration of the curve number method (CN) from data. The focus of the study is understanding the variation of CN as a function of precipitation by analysing the variability of initial infiltration over the catchment. I particularly liked the analysis of the inconsistency of the theoretical definition of initial abstraction (Ia) and its value at the watershed scale. Based on their analysis, the authors propose a set of models with increasing complexity. They apply these models to synthetic basins with controlled heterogeneity following the CN behaviour and compare their performance with standard indices. By introducing additional parameters in the CN method they obtain a good fit of the precipitation-runoff relationship resulting from the application of the CN method

to heterogeneous basins.

The topic is relevant for the audience of Hydrology and Earth System Science, because the CN is the most widely used method to account for infiltration losses in professional applied hydrology. The objectives of the study are clearly identified, the methodology for the analysis is sound and the conclusions are relevant and correctly supported by the results and discussion. The proposed models perform well when reproducing the behaviour of heterogeneous basins and there are reasonable expectations that the method can be applied to natural basins. Therefore, I believe the paper deserves publication in Hydrology and Earth System Science.

SPECIFIC COMMENTS I also think that there are several aspects of the paper that deserve a deeper discussion, such as the following:

a) On page 2, lines 8-9, the authors state that, in addition to varying spatially due to watershed heterogeneity, CN also varies temporally due to changes in soil moisture or vegetation cover. However, in their synthetic experiments they did not account for temporal variation of CN or Ia. In my opinion, this is a significant limitation for the practical application of the proposed models, that were tested under steady conditions. The authors cite a forthcoming paper by themselves (Santikari and Murdoch, 2018) where several ways of dealing with temporal variation of CN are proposed. I was not able to locate such paper in HESSD. I think this issue should be briefly discussed in this paper, maybe in a section devoted to the limitations of the methodology presented here.

b) The promised paper (Santikari and Murdoch, 2018) will also deal with application to real watersheds, not synthetic data (lines 23-26). The argument given in favour of using synthetic watersheds (page 30, lines 12-16) is sound. However, the strength of the proposed CN-based methods lies on their practical applications. Since the authors have analysed applications to real watersheds, I think a brief discussed of this issue should also be included in this paper.

c) On page 11, lines 17 to 21, the authors report the standard professional practice of accounting for heterogeneity by obtaining the area-weighted average of the CN. The results presented in the paper clearly show that this practice can be improved. I think the authors should discuss this in the final part of the paper. This practice is routinely applied in ungauged basins, where CN is estimated from physiographic characteristics. Are there any better alternatives for computing an average CN in view of the research carried out? Can they propose a model for ungauged basins? I am aware this is not the main objective of the work, but I think the paper would benefit from a discussion of this issue.

d) The models were tested just for one synthetic watershed (described in table 2). This is a limitation of the methodology. The comparative results of model performance would certainly depend on the degree of heterogeneity of the tested basin. In suggest that a discussion of this issue be included in the paper and acknowledged in the conclusions.

TECHNICAL CORRECTION From the formal standpoint, the paper is very well written, correctly organized and adequately illustrated with tables and figures. Figures 8 and 9 could benefit from the use of colours, if possible. Although I am not a native English speaker, I believe the following expression should be corrected:

On page 32, line 12 and 14, . . . in both the cases...... ("the" should be removed?).

---

## Author Comment (AC1) · 15 Jan 2018

We thank the referee for reviewing our manuscript and for providing valuable suggestions. Our responses to specific comments are given below. Please refer to the list of symbols in the paper as needed.
* * *
GENERAL COMMENTS The authors present a methodology to account for heterogeneity in the calibration of the curve number method (CN) from data. The focus of the study is understanding the variation of CN as a function of precipitation by analyzing the variability of initial infiltration over the catchment. I particularly liked the analysis of the inconsistency of the theoretical definition of initial abstraction (Ia) and its value at the watershed scale. Based on their analysis, the authors propose a set of models with increasing complexity. They apply these models to synthetic basins with controlled heterogeneity following the CN behaviour and compare their performance with standard indices. By introducing additional parameters in the CN method they obtain a good fit of the precipitation-runoff relationship resulting from the application of the CN method to heterogeneous basins.

The topic is relevant for the audience of Hydrology and Earth System Science, because the CN is the most widely used method to account for infiltration losses in professional applied hydrology. The objectives of the study are clearly identified, the methodology for the analysis is sound and the conclusions are relevant and correctly supported by the results and discussion. The proposed models perform well when reproducing the behaviour of heterogeneous basins and there are reasonable expectations that the method can be applied to natural basins. Therefore, I believe the paper deserves publication in Hydrology and Earth System Science.

SPECIFIC COMMENTS I also think that there are several aspects of the paper that deserve a deeper discussion, such as the following:
* * *
**Referee's Comment**

a) On page 2, lines 8-9, the authors state that, in addition to varying spatially due to watershed heterogeneity, CN also varies temporally due to changes in soil moisture or vegetation cover. However, in their synthetic experiments they did not account for temporal variation of CN or Ia. In my opinion, this is a significant limitation for the practical application of the proposed models, that were tested under steady conditions. The authors cite a forthcoming paper by themselves (Santikari and Murdoch, 2018) where several ways of dealing with temporal variation of CN are proposed. I was not able to locate such paper in HESSD. I think this issue should be briefly discussed in this paper, maybe in a section devoted to the limitations of the methodology presented here.

**Authors' Response**

We agree that not accounting for temporal variations is a limitation of the models proposed. This is why we extended our work to develop models that account for temporal variations, and we present them in the companion paper (Santikari and Murdoch, 2018), which is under review in Water Resources Management. We provided that manuscript during our original submission of this paper to HESS as "author's response", but it appears that it was not accessible. We would like to give the referees access to the companion manuscript, but we are unaware of how to do so. We apologize for the inconvenience.

The focus of the current paper is the inclusion of spatial variations. We first show that the poorly understood variation of CN with $P$ is due to watershed heterogeneity. Then we propose models that treat the CN method's parameters as functions of $P$, to account for spatial variations and improve runoff predictions. Although accounting for temporal variations further improves runoff predictions (Santikari and Murdoch, 2018), the proposed models in the current paper have two advantages: (i) they are simpler, and (ii) their data requirements are lower. Models that account for temporal variations are relatively more complex, and they require continuous rainfall-runoff observations for calibration (Santikari and Murdoch, 2018). So, the models proposed in this paper are applicable when continuous observations are absent, and they may be preferable in some cases because of their simplicity.

As per the referee's suggestion, we added a subsection to state the model limitation as follows:

**6.5. Model Limitation**

A strength of the models proposed in this paper is that they provide a compact way to account for the spatial variation of CN, $I_a$, or $S$ (watershed heterogeneity), but a limitation is that they do not account for the temporal variation. During dry periods $I_a$, and $S$ increase whereas CN decreases. The behavior is opposite during the wet periods. Changes in land cover introduce additional temporal variations. Therefore, the calibrated model parameters in this paper can be considered as temporal averages. The models may underpredict runoff during wet periods and overpredict during dry periods. A procedure to account for temporal variations using antecedent moisture is described in the companion paper (Santikari and Murdoch, 2018).
* * *
**Referee's Comment**

b) The promised paper (Santikari and Murdoch, 2018) will also deal with application to real watersheds, not synthetic data (lines 23-26). The argument given in favour of using synthetic watersheds (page 30, lines 12-16) is sound. However, the strength of the proposed CN-based methods lies on their practical applications. Since the authors have analysed applications to real watersheds, I think a brief discussed of this issue should also be included in this paper.

**Authors' Response**

As per referee's suggestion, we added a subsection to briefly discuss the results from the application of the proposed models to real watersheds as follows:

**6.2. Application to Real Watersheds**

The models were also evaluated using rainfall-runoff observations from 9 real watersheds located in different parts of the world (Santikari and Murdoch, 2018). Models' ability to predict the observed runoff was assessed using $NSE_Q$. Results show that in all the watersheds VIMs performed better than CMs but

the difference in performance, $\Delta NSE_Q$, varied across the watersheds. Between VIMλ and CM0.2, $\Delta NSE_Q$ < 0.05 in one watershed, $0.05 \leq \Delta NSE_Q < 0.7$ in 6 watersheds, and $\Delta NSE_Q \geq 0.7$ in 2 watersheds. Between VIMλ and CMλ, $\Delta NSE_Q < 0.05$ in 3 watersheds, $0.05 \leq \Delta NSE_Q < 0.1$ in 4 watersheds, and $\Delta NSE_Q \geq 0.1$ in 2 watersheds. Based on their performance, the models can be arranged from the best to the worst as VIMλ > VIMS > CMλ > CM0.2, which is consistent with results from their application to the synthetic watershed.

**Note:** The current subsection "6.2. Model Suitability" has been moved to 6.4.
* * *
**Referee's Comment**

c) On page 11, lines 17 to 21, the authors report the standard professional practice of accounting for heterogeneity by obtaining the area-weighted average of the CN. The results presented in the paper clearly show that this practice can be improved. I think the authors should discuss this in the final part of the paper. This practice is routinely applied in ungauged basins, where CN is estimated from physiographic characteristics. Are there any better alternatives for computing an average CN in view of the research carried out? Can they propose a model for ungauged basins? I am aware this is not the main objective of the work, but I think the paper would benefit from a discussion of this issue.

**Authors' Response**

We thank the referee for raising this important issue. The alternative to using an average CN is to use an average $Q$, i.e. calculate runoff from each HRU and take the area-weighted average to get the runoff from the watershed. This procedure accounts for heterogeneity (the spatial variation of CN). If watershed scale Ia and CN are estimated from this area-weighted $Q$, they will vary with $P$ as shown in Figure 2. Averaging CN is easier to use but averaging $Q$ is more accurate.

Throughout the paper, we referred to this approach of averaging $Q$ as distributed parameter CN model. We acknowledged in the introduction (page 2, lines 3 and 4) that this approach can account for

heterogeneity when it is known at sufficient detail. In section 2.1.1, we mentioned that averaging $Q$ is a better approach than averaging CN (page 9, lines 8 to 11). We also used this approach to generate synthetic runoff (Section 5.1: page 23, lines 5 to 7), which was used in the evaluation of the lumped parameter models. We feel that the information provided in the paper makes a strong case in favor of averaging $Q$ over averaging CN.
* * *
**Referee's Comment**

d) The models were tested just for one synthetic watershed (described in table 2). This is a limitation of the methodology. The comparative results of model performance would certainly depend on the degree of heterogeneity of the tested basin. In suggest that a discussion of this issue be included in the paper and acknowledged in the conclusions.

**Authors' Response**

Although we presented results for one synthetic watershed, we tested the models for several distributions of heterogeneity. The summary of our findings was that CM0.2 was always the worst whereas VIMs were better or equal in performance to CMλ. As the referee correctly pointed out, the difference in performance between VIMs and CMλ depends on the degree of heterogeneity. To illustrate this, we evaluated the models for a different distribution of heterogeneity and presented the results in a new subsection as follows.

**6.3. Effect of Degree of Heterogeneity**

The degree of heterogeneity, defined as the sharpness of change in CN, $I_a$, or $S$ between the HRUs, may affect the relative performance of the models. To verify this, the degree of heterogeneity of the synthetic watershed (Table 2) was increased by doubling the $S_i$s for HRUs 3 and 4 while the others were left unchanged, i.e. the modified distribution was $S_0 = 0$ mm, $S_1 = 50$ mm, $S_2 = 100$ mm, $S_3 = 300$ mm, and $S_4 = 400$ mm. The models were applied to this modified synthetic watershed, for the cases of $\lambda_i = 0.2$ and 0.5, and their performances were assessed using $SEE_Q$.

Comparing the results (Tables 3 and 4) shows that the performance of VIMs remained nearly the same, whereas the performance of CM0.2 decreased and that of CMλ increased. The relative order of performance remained unchanged, i.e. VIMλ > VIMS > CMλ > CM0.2.

**Table 4.** Performance of the models for the cases of $\lambda_i = 0.2$ and 0.5, assessed using SEE$_Q$, when the degree of heterogeneity in the synthetic watershed (Table 2) was increased by doubling the $S_i$s for HRUs 3 and 4.

| Model | $\lambda_i = 0.2$ | $\lambda_i = 0.5$ |
|---|---|---|
| CM0.2 | 1.54 | 1.30 |
| CMλ | 0.19 | 0.38 |
| VIMS | 0.12 | 0.25 |
| VIMλ | 0.06 | 0.12 |

The results from real watersheds (Santikari and Murdoch, 2018) also show that the performance of CM0.2 was poor, NSE$_Q$ < 0.25, in watersheds with a sharp change in CN. Therefore, CM0.2 appears to be unsuitable when the degree of heterogeneity is large. CMλ performed moderately well on synthetic and real watersheds with a large degree of heterogeneity, possibly by transferring the storage (Section 6.1). So, CMλ is suitable for predicting overall runoff, but less reliable for predicting heterogeneity or runoff from small events. VIMs outperformed CMλ in synthetic (Table 4) as well as real watersheds (Santikari and Murdoch, 2018) with a large degree of heterogeneity, and therefore they are more reliable.

**Note:** The results in Table 4 were obtained from a synthetic watershed that is different than the original (Table 2). Similar results can be obtained by modifying the HRU distribution in Table 2 and reapplying the models. We chose not to present results from several synthetic watersheds because of the similarity of the results (i.e. same relative model performance) and space limitations.

**Referee's Comment**

TECHNICAL CORRECTION From the formal standpoint, the paper is very well written, correctly organized and adequately illustrated with tables and figures. Figures 8 and 9 could benefit from the use of colours, if possible. Although I am not a native English speaker, I believe the following expression should be corrected: On page 32, line 12 and 14, in both the cases...... ("the" should be removed?).

**Authors' Response**

We appreciate the comments on the writing and organization in our paper. We agree that figures 8 and 9 would benefit from color, and we used colors in the original submission. It appears that the file that was reviewed may have been converted to black and white. We also noted that the line and page numbers used by the referee are different from those in the copy we submitted, which is same as the pdf file that can be currently downloaded from the HESS website. We are unsure of how and when the file conversion has occurred. The line and page numbers we used in this response correspond to the pdf file that can be currently downloaded. We apologize for the inconvenience.

We thank the referee for pointing out the grammatical error, "the" has been removed.

---

## Referee Comment (RC2) · Anonymous Referee #2 · 23 Jan 2018

GENERAL COMMENTS

This manuscript aims at improving the SCS-CN method and the estimation of the corresponding parameters when rainfall runoff data are available. The proposed approach is based on the hypothesis posed by Soulis and Valiantzas (2012) that "the observed correlation between the calculated CN value and the rainfall depth in a watershed reflects the effect of the inevitable presence of soil-cover complex spatial variability along watersheds". Based on this hypothesis they present a novel and really interesting analysis of the effects of this heterogeneity on initial abstraction and on CN. It includes nice theoretical justifications, and good examples. In a nutshell, their analysis provides

another more general perspective extending the work of Soulis and Valiantzas (2012, 2013) by considering separately the spatial variability of Ia and CN, which are linked in the previous studies. Finally, based on their analysis they introduce two modifications of the SCS-CN method considering the spatial variability.

The topic of this study is certainly interesting and relevant to the journal of Hydrology and Earth System Science, because the SCS-CN is the most widely used runoff estimation method, while it is based on previous studies published at this journal. The study is very well written and really easy to understand. The language is excellent and the presentation also of good quality. The theoretical part is also interesting and well written and the interpretations and the methodology scientifically sound.

However, there are also some important weaknesses that should be addressed.

The first important weakness is related to the citation of an unpublished paper. The citation of studies that are not published yet and thus are not available to the readers isn't helpful. This is not a significant problem at the first instance (Page 2, Line 6), where there is a general reference on "ways to account for the temporal variation of CN, each with its own advantages and shortcomings (Santikari and Murdoch, 2018)". In this instance the citation on the unpublished work should be removed and some citations on studies dealing with this issue should be added. However, in the second instance (Page 23, Lines 14-15) an unpublished paper is used to support the validity of the proposed approach and the performance of the proposed modifications ["Application of these modified models to data from real watersheds is discussed by Santikari and Murdoch (2018)"]. Any information concerning real watersheds examples should be presented in this paper (the part related to the proposed approach). Otherwise the readers will not be able to have a clear picture about the validity and the performance of the proposed approach. Furthermore, there are practical problems in citing unpublished papers. Are you certain that the future paper will be accepted and that it is going to be published before the final publication of this paper?

A second weakness, is related to the use of solely synthetic data for the evaluation of the proposed approach and of the proposed modifications. (Page 23, Lines 12-14: "The reason for using a synthetic watershed here is that the heterogeneity can be precisely defined and used to evaluate the predictions of heterogeneity by the lumped parameter models. In real watersheds the heterogeneity has to be determined by calibration, and there can be non-uniqueness when multiple HRUs are present.") I agree that using a virtual watershed allows the study and the evaluation of specific aspects of your approach in a controlled and accurate environment. The virtual watershed and the synthetic data follow the logic of your base hypothesis and your theoretical analysis. However, this hypothesis and this analysis, even if they are rational, they are not self-evident. The reason for using also real watersheds examples is that only in this way you may show that your hypothesis is sound, that it is able to describe the behaviour of real watersheds, and that the method actually works. By using only virtual data generated based on your hypothesis (which, I agree, seems reasonable) you cannot support your hypothesis and validate your methodology.

A final weakness concerns the literature review, which is limited and incomplete. For example:

1. Page 1, Lines 26-28: "One of the most popular techniques used for this purpose is the Curve Number method, which has been in use for more than half a century (Soil Conservation Service, 1956)." You should add some citations supporting this statement.

2. Page 2, Lines 2-3: "CN also varies with the magnitude and spatiotemporal distribution of rainfall." You should add some citations supporting this statement.

3. Page 2, Lines 3-5: "When heterogeneity is known at sufficient detail, CN variation can be accounted by using a distributed parameter model. Otherwise this approach can introduce more parameters than can be reliably estimated from the available data, and cause large uncertainties 5 in the predicted runoff." You should add the citations

supporting this statement, for example Soulis and Valiantzas (2012, 2013) referred later in the manuscript.

4. Page 2, Lines 5-9: "CN variation with the distribution of rainfall is usually ignored." and "CN method is most commonly applied as an event-scale lumped parameter model, which is simple but also limited in its ability to account for the variations of CN. This diminishes the accuracy of its runoff predictions." You should add some citations supporting these two statements (E.g. Grove et al. (1998); Soulis and Valianzas, 2012).

5. Page 4, Lines 26: There more studies providing important information on this issue e.g. Hjelmfeld et al. (2001) and Soulis et al., (2009)

6. Page 7, Lines 10-17: the studies of Soulis and Valianzas, (2012, 2013) should be also mentioned at this point.

More important, you should consider previous studies dealing with the same issue with similar or different approaches. I have in mind for example two really important studies by Steenhuis et al., 1995 and its continuation by Tilahun et al., 2016 that investigate the variation of Ia using the concept of "Variable source runoff areas" and propose a very attractive approach to consider it in the SCS-CN method. You should discuss these studies.

You should also state more clearly that the proposed approach is based on the hypothesis posed by Soulis and Valiantzas (2012) that "the observed correlation between the calculated CN value and the rainfall depth in a watershed reflects the effect of the inevitable presence of soil-cover complex spatial variability along watersheds". You should also make it clear and add a citation to Soulis and Valiantzas (2012, 2013) in Page 15, Lines 9-10: "Therefore, it is probably more appropriate to refer to any "CN decreasing with P" trend as standard behavior, because it is caused by the inevitable presence of heterogeneity in a watershed."

Finally, you should discuss your results in comparison with other approaches/methods especially at the final section "6.2. Model Suitability". You should also mention other limitations such as the compatibility of the resulted CN values with standard method. For example, CN values with different $\lambda$ values are not compatible.

SPECIFIC COMMENTS

-Page 5, Line 28: You should mention what is presented in the figure.

-Figure 2, legend: "(see Santikari and Murdoch (2018) for study area description)" You should avoid citing unpublished work (see previous comments). You should provide at least a short description of the case study.

-Page 7, Lines 23-24: How Ia values in Figure 2 were calculated?

-Please avoid using plural in parameters symbols. For example, in "Iais" I was initially confused if s was for plural or part of the symbol. You may use other explosion such as "Iai values".

-Page 9, Lines 9-11: As it is explained in Soulis and Valiantzas (2012), the reason is the non-linear form of the SCS-CN formula. So, the average of the results is not equal with the result using average value of the parameters.

-Page 9, Line 13: "to be"

-Page 9, "2.2. Ia in a Heterogeneous Watershed": It should be mentioned that the following justification is valid in the case that each subarea is directly connected with the drainage network. This is a logical assumption in most cases, especially when there is a dense drainage network, however, it is still an assumption.

-Page 34, Lines 9-14: You could use additional evaluation criteria e.g. the relative NSE (rNSE) and the NSE with logarithmic values (lnNSE) to reduce the problem of the NSE sensitivity to extreme values (see Krause et al., 2005).

-"Model Suitability" section: It would be interesting if you could at least discuss (if it

is not possible to compare) with the Soulis and Valiantzas 2012 and 2013 methods, which provided the base for this study.

Conclusively, based on the above comments, I believe that this paper is really interesting and worth being published in case that the authors are able to address the above issues.

REFERENCES

Grove, M., Harbor, J., and Engel, B.: Composite vs. Distributed curve numbers: Effects on estimates of storm runoff depths, J. Am. Water Resour. As., 34, 1015–1023, 1998.

Hjelmfelt Jr., A. T., Woodward, D. A., Conaway, G., Plummer, A., Quan, Q. D., Van Mullen, J., Hawkins, R. H., and Rietz, D.: Curve numbers, recent developments, in: Proc. of the 29th Congress of the Int. As. for Hydraul. Res., Beijing, China (CD ROM), 17–21 September, 2001.

Krause, P., Boyle, D.P., Base, F., 2005. Comparison of different efficiency criteria for hydrological model assessment. Adv. Geosci. 5, 89–97.

Soulis, K. X., Valiantzas, J. D., Dercas, N., and Londra P. A.: Analysis of the runoff generation mechanism for the investigation of the SCS-CN method applicability to a partial area experimental watershed, Hydrol. Earth Syst. Sc. 13, 605–615, doi:10.5194/hess-13-605-2009, 2009.

Soulis K.X. and Valiantzas J.D. 2012. SCS-CN parameter determination using rainfall-runoff data in heterogeneous watersheds - the two-CN system approach. Hydrol. Earth Syst. Sci. 16:1001-1015. doi: 10.5194/hess-16-1001-2012.

Soulis K.X. and Valiantzas J.D. 2013. Identification of the SCS-CN parameter spatial distribution using rainfall-runoff data in heterogeneous watersheds. Water Resour. Manage. 27:1737-1749. doi: 10.1007/s11269-012-0082-5.

Steenhuis, T.S., Winchell, M., Rossing, J., Zollweg, J.A., and Walter, M.F. (1995). SCS

runoff equation revisited for variable-source runoff areas, J. Irrig. Drain. Eng. ASCE, 121, 234–238,.

Tilahun, S.A., Ayana, E.K., Guzman, C.D., Dagnew, D.C., Zegeye, A.D., Tebebu, T.Y., Steenhuis, T.S. (2016). Revisiting storm runoff processes in the upper Blue Nile basin: The Debre Mawi watershed. Catena, 143, 47-56. doi:10.1016/j.catena.2016.03.029

---

## Author Comment (AC2) · 19 Mar 2018

We thank the referee for their time, and for their detailed comments and suggestions for our manuscript. Our responses, wherever appropriate, are given below.

GENERAL COMMENTS

This manuscript aims at improving the SCS-CN method and the estimation of the corresponding parameters when rainfall runoff data are available. The proposed approach is based on the hypothesis posed by Soulis and Valiantzas (2012) that "the observed correlation between the calculated CN value and the rainfall depth in a watershed reflects the effect of the inevitable presence of soil-cover complex spatial variability along watersheds". Based on this hypothesis they present a novel and really interesting analysis of the effects of this heterogeneity on initial abstraction and on CN. It includes nice theoretical justifications, and good examples. In a nutshell, their analysis provides another more general perspective extending the work of Soulis and Valiantzas (2012, 2013) by considering separately the spatial variability of Ia and CN, which are linked in the previous studies. Finally, based on their analysis they introduce two modifications of the SCS-CN method considering the spatial variability.

The topic of this study is certainly interesting and relevant to the journal of Hydrology and Earth System Science, because the SCS-CN is the most widely used runoff estimation method, while it is based on previous studies published at this journal. The study is very well written and really easy to understand. The language is excellent and the presentation also of good quality. The theoretical part is also interesting and well written and the interpretations and the methodology scientifically sound.

**Referee's Comment**

However, there are also some important weaknesses that should be addressed. The first important weakness is related to the citation of an unpublished paper. The citation of studies that are not published yet and thus are not available to the readers isn't helpful. This is not a significant problem at

the first instance (Page 2, Line 6), where there is a general reference on "ways to account for the temporal variation of CN, each with its own advantages and shortcomings (Santikari and Murdoch, 2018)". In this instance the citation on the unpublished work should be removed and some citations on studies dealing with this issue should be added. However, in the second instance (Page 23, Lines 14-15) an unpublished paper is used to support the validity of the proposed approach and the performance of the proposed modifications ["Application of these modified models to data from real watersheds is discussed by Santikari and Murdoch (2018)"]. Any information concerning real watersheds examples should be presented in this paper (the part related to the proposed approach). Otherwise the readers will not be able to have a clear picture about the validity and the performance of the proposed approach. Furthermore, there are practical problems in citing unpublished papers. Are you certain that the future paper will be accepted and that it is going to be published before the final publication of this paper?

**Authors' Response**

We understand the referee's concern about the citation of the companion paper which is under review and therefore unavailable to the readers. To overcome this problem, we now cited my Ph.D. dissertation (Santikari, 2017) wherever a citation to the companion paper appears. My dissertation is published and easily accessible to the readers, and it has a chapter that is identical to the companion paper. We also provided a link to the dissertation under references in this response, and we hope it will also be helpful to the referees.

We also feel that the citations to the companion paper should be left in place. The work presented in the companion paper is a logical extension to the work presented in the current paper, so it is beneficial to the readers to be aware of that. However, we will update the reference to the companion paper as "submitted", "under review", or "accepted" based on its status before the final publication of the current paper.

Referee #1 also suggested describing the results from the application of the models to real watersheds. We now added a subsection to briefly discuss these results as follows (copied from Response to Referee #1):

**6.2. Application to Real Watersheds**

The models were also evaluated using rainfall-runoff observations from 9 real watersheds located in different parts of the world (Santikari, 2017; Santikari and Murdoch, 2018). Models' ability to predict the observed runoff was assessed using $NSE_Q$. Results show that in all the watersheds VIMs performed better than CMs but the difference in performance, $\Delta NSE_Q$, varied across the watersheds. Between VIM$\lambda$ and CM0.2, $\Delta NSE_Q < 0.05$ in one watershed, $0.05 \leq \Delta NSE_Q < 0.7$ in 6 watersheds, and $\Delta NSE_Q \geq 0.7$ in 2 watersheds. Between VIM$\lambda$ and CM$\lambda$, $\Delta NSE_Q < 0.05$ in 3 watersheds, $0.05 \leq \Delta NSE_Q < 0.1$ in 4 watersheds, and $\Delta NSE_Q \geq 0.1$ in 2 watersheds. Based on their performance, the models can be arranged from the best to the worst as VIM$\lambda$ > VIMS > CM$\lambda$ > CM0.2, which is consistent with results from their application to the synthetic watershed.

A detailed description of the study areas, application, and results (Santikari, 2017) is impractical because that would make the current paper significantly lengthy. The readers can refer to Santikari (2017), which describes the case studies in more detail than could be presented in the current paper.

The current paper, even without a description of application to real watersheds, makes an important contribution to our understanding of CN methodology by providing a theoretical proof that heterogeneity leads to the variation of CN parameters with $P$. The paper also throws light on the long-known but unexplained phenomenon of the calibrated $\lambda$ being much smaller than 0.2, and introduces the novel concept of storage transfer.
* * *
**Referee's Comment**

A second weakness, is related to the use of solely synthetic data for the evaluation of the proposed approach and of the proposed modifications. (Page 23, Lines 12-14: "The reason for using a synthetic

watershed here is that the heterogeneity can be precisely defined and used to evaluate the predictions of heterogeneity by the lumped parameter models. In real watersheds the heterogeneity has to be determined by calibration, and there can be non-uniqueness when multiple HRUs are present.") I agree that using a virtual watershed allows the study and the evaluation of specific aspects of your approach in a controlled and accurate environment. The virtual watershed and the synthetic data follow the logic of your base hypothesis and your theoretical analysis. However, this hypothesis and this analysis, even if they are rational, they are not selfevident. The reason for using also real watersheds examples is that only in this way you may show that your hypothesis is sound, that it is able to describe the behaviour of real watersheds, and that the method actually works. By using only virtual data generated based on your hypothesis (which, I agree, seems reasonable) you cannot support your hypothesis and validate your methodology.

**Authors' Response**

We agree that ultimately the new methodology is judged by whether it describes the behavior of real watersheds, and by whether it works i.e. whether it produces better results than the conventional models when applied to real watersheds. The paper in its current form shows that the new models meet both criteria.

1. The motivation for deriving the new methodology are the observations from real watersheds in our study (Figure 2), and from real watersheds in other studies (D'Asaro and Grillone 2012; Hawkins, 1993), which show that CN variation with $P$ is common. So the variation of $I_a$, which is inversely related to CN (eq. 10), with $P$ is also a commonly observed behavior in real watersheds. Therefore, the new models which allow for the variation of $I_a$ with $P$ describe the behavior of real watersheds better than the conventional models, which assume that $I_a$ is constant.

2. The results from real watersheds, summarized in the newly added Subsection 6.2, show that the new models predict runoff better than the conventional models. This proves that the new

methodology works better even with real watersheds.
* * *
A final weakness concerns the literature review, which is limited and incomplete. For example:

1. Page 1, Lines 26-28: "One of the most popular techniques used for this purpose is the Curve Number method, which has been in use for more than half a century (Soil Conservation Service, 1956)." You should add some citations supporting this statement.

**Authors' Response**

The following citations have been added:

D'Asaro and Grillone, 2012; Hawkins et al., 2008; Kent, 1968; Ponce and Hawkins, 1996; Rallison and Miller, 1982; Soil Conservation Service, 1972
* * *
2. Page 2, Lines 2-3: "CN also varies with the magnitude and spatiotemporal distribution of rainfall." You should add some citations supporting this statement.

**Authors' Response**

Citations have been added as follows:

CN also varies with the magnitude (D'Asaro and Grillone, 2012; Hawkins, 1993; Hjelmfelt et al., 2001) and spatiotemporal distribution of rainfall (Hawkins et al., 2008; Van Mullem, 1997).
* * *
3. Page 2, Lines 3-5: "When heterogeneity is known at sufficient detail, CN variation can be accounted by using a distributed parameter model. Otherwise this approach can introduce more parameters than can be reliably estimated from the available data, and cause large uncertainties 5 in the predicted runoff." You should add the citations supporting this statement, for example Soulis and Valiantzas (2012, 2013) referred later in the manuscript.

**Authors' Response**

Citations have been added as follows:

When heterogeneity is known at sufficient detail, CN variation can be accounted by using a distributed parameter model, e.g. SWAT (Gassman et al., 2007). Otherwise this approach can introduce more parameters than can be reliably estimated from the available data (Soulis and Valiantzas, 2013), and can potentially cause large uncertainties in the predicted runoff.
* * *
**Referee's Comment**

4. Page 2, Lines 5-9: "CN variation with the distribution of rainfall is usually ignored." and "CN method is most commonly applied as an event-scale lumped parameter model, which is simple but also limited in its ability to account for the variations of CN. This diminishes the accuracy of its runoff predictions." You should add some citations supporting these two statements (E.g. Grove et al. (1998); Soulis and Valianzas, 2012).

**Authors' Response**

Citations have been added as follows:

CN variation with the distribution of rainfall is usually ignored (Hawkins et al., 2008). CN method is most commonly applied as an event-scale lumped parameter model, which is simple but also limited in its ability to account for the variations of CN. This diminishes the accuracy of its runoff predictions (e.g.

Soulis and Valianzas, 2012).
* * *
5. Page 4, Lines 26: There more studies providing important information on this issue e.g. Hjelmfeld et al. (2001) and Soulis et al., (2009)

**Authors' Response**

We added a sentence at Page 4, Line 26 as:

This behavior was also observed in several previous studies (D'Asaro and Grillone, 2012; Hawkins, 1993; Hjelmfelt et al., 2001; Soulis et al., 2009), and it appears to be a common phenomenon. Hawkins (1993) and D'Asaro and Grillone (2012) evaluated approximately 100 watersheds in a wide range of settings, and in 75% of the watersheds they observed…
* * *
6. Page 7, Lines 10-17: the studies of Soulis and Valianzas, (2012, 2013) should be also mentioned at this point.

**Authors' Response**

Citations have been added for the following sentences:

This is because when CN is constant, $Q$ may be underestimated for small $P$ and overestimated for large $P$ [e.g. Soulis and Valianzas (2012, 2013)]. This problem can be addressed either by treating CN as a function of $P$, e.g. asymptotic fitting (Hawkins, 1993), or by using a distributed modeling approach that accounts for heterogeneity in sufficient detail, e.g. SWAT (Gassman et al., 2007) or Soulis and Valianzas (2013).
* * *
**Referee's Comment**

More important, you should consider previous studies dealing with the same issue with similar or different approaches. I have in mind for example two really important studies by Steenhuis et al., 1995 and its continuation by Tilahun et al., 2016 that investigate the variation of Ia using the concept of "Variable source runoff areas" and propose a very attractive approach to consider it in the SCS-CN method. You should discuss these studies.

**Authors' Response**

Although Steenhuis et al. (1995) present an interesting interpretation of the CN method, we think that their approach is fundamentally different to ours, and falls beyond the scope of the topic covered in our manuscript for the following reasons:

1. A fundamental assumption made in the approach used by Steenhuis et al. (1995) is that for a given patch of land, $dQ/dP_e$ is either zero or unity, i.e. either it produces no runoff or 100% of the rainfall becomes runoff. According to this assumption, Figure 1 in our manuscript would look like a step function. However, the assumption in the original CN method, as well as in our manuscript, is that $dQ/dP_e$ increases with $P_e$ and varies as $0 \leq dQ/dP_e \leq 1$. The assumption by Steenhuis et al. (1995) is applicable only in watersheds with variable source runoff areas.

2. At the watershed scale, however, $dQ/dP_e$ is allowed to vary between zero and unity in the approach used by Steenhuis et al. (1995). This is a result of recasting the definition of $S$ from being parameter that describes how the vertical storage is the soil profile is filled to a parameter that describes how saturated areas develop in the horizontal dimension. In other words, in the original CN method, $S$ describes how $F$ or $Q$ vary with $P$ at a given location or for an entire watershed. In the method by Steenhuis et al. (1995), $S$ describes how the areal extent of saturation and the resultant $Q$ vary with $P$ only at the watershed scale.

3. It is impossible to handle watershed heterogeneity using the method by Steenhuis et al. (1995). This is because in each HRU, $I_{ai} \geq 0$ but $F_i = 0$ and $S_i = 0$. This stems from the fundamental assumption of their method (see point 1 above). Before the onset of runoff, $P < I_{ai}$, i.e. the initial abstraction has not yet been overcome. But as soon as the runoff starts, all the additional rainfall becomes runoff, and this can only happen if $S_i = 0$. Thus, $S$ only exists at the watershed scale but vanishes at the HRU scale. Moreover, in their method $I_a$ and $S$ are constants at the watershed scale for given set of antecedent conditions. The topic covered our manuscript, however, is focused on how to include heterogeneity in the CN method, and in our approach $I_a$ and $S$ vary spatially and with $P$.
* * *
**Referee's Comment**

You should also state more clearly that the proposed approach is based on the hypothesis posed by Soulis and Valiantzas (2012) that "the observed correlation between the calculated CN value and the rainfall depth in a watershed reflects the effect of the inevitable presence of soil-cover complex spatial variability along watersheds".

**Authors' Response**

We would like to clarify here that when we discovered the standard behavior in our watersheds (Figure 2) and sought an explanation, we were unaware of the work done by Soulis and Valiantzas (2012, 2013). Our analysis followed the logic described in Section-2 and lead to the theoretical proof that heterogeneity causes standard behavior. We came across the work of Soulis and Valiantzas (2012) long after we finished our analysis, and we were surprised to see that they reached the same conclusion using an empirical analysis. We did give them the credit in the introduction because they were the first to discover the link between heterogeneity and standard behavior. However, it would be inaccurate for us to say that our approach was based on the hypothesis by Soulis and Valiantzas (2012, 2013).
* * *
**Referee's Comment**

You should also make it clear and add a citation to Soulis and Valiantzas (2012, 2013) in Page 15, Lines 9-10: "Therefore, it is probably more appropriate to refer to any "CN decreasing with P" trend as standard behavior, because it is caused by the inevitable presence of heterogeneity in a watershed."

**Authors' Response**

To be accurate with the citations, the above sentence was split and recast as follows:

…supporting the hypothesis that $I_{aW} = I_{aF}$. Thus, as also concluded by Soulis and Valiantzas (2012, 2013), the observed *complacent* and *standard behaviors* are caused by the inevitable presence of heterogeneity in a watershed. Moreover, *complacent behavior* appears to be a special case of *standard behavior* (Soulis and Valiantzas, 2012), where observations from larger rainfalls are unavailable. Therefore, it is probably more appropriate to refer to any "CN decreasing with $P$" trend as *standard behavior*.
* * *
**Referee's Comment**

Finally, you should discuss your results in comparison with other approaches/methods especially at the final section "6.2. Model Suitability".

**Authors' Response**

We compared variable $I_a$ models with conventional lumped parameter models in the results from both synthetic and real watersheds. Distributed parameter modeling is another approach that could be compared with modified models. As per referee's suggestion, we briefly discussed this under "6.4. Model Suitability" as:

When the watershed heterogeneity is known in great detail such that the number of calibrated parameters ≤ 3, a distributed modeling approach [e.g. SWAT (Gassman et al., 2007) or Soulis and Valianzas (2013)] may be preferable over the variable $I_a$ models. A distributed parameter model has advantages similar to

the variable $I_a$ models over the conventional models. It would inherently account for the variation of CN method's parameters spatially and with $P$. It would also avoid the false prediction of zero runoffs in small events because HRUs with larger CNs, which generate runoff even in small events, are explicitly considered. When the heterogeneity is unknown, however, the number of calibrated parameters (for values of $CN_i$ and $a_i$) in a distributed model with $n$ HRUs is $2n-1$. This number would increase further if values of $\lambda_i$ are also calibrated. Therefore, when the number of calibrated parameters $> 3$, application of a variable $I_a$ model should be considered.
* * *
**Referee's Comment**

You should also mention other limitations such as the compatibility of the resulted CN values with standard method. For example, CN values with different $\lambda$ values are not compatible.

**Authors' Response**

We added the following paragraph to "6.5. Model Limitations"

Another limitation of VIMs is that the CN values calculated using eqs. (5) or (10) are incompatible with the standard CN values (NRCS, 2003; USDA, 1986) derived using CM0.2. However, this limitation is not unique to VIMs because any method, including CM$\lambda$, which involves an altered relationship between $I_a$ and $S$ (i.e. $\lambda \neq 0.2$) leads to CN values that are incompatible with those derived from CM0.2. Given that (i) CM0.2 is a poor predictor of runoff, and (ii) the evidence contradicts $\lambda = 0.2$ (Baltas et al., 2007; D'Asaro and Grillone, 2012; Shi et al., 2009; Woodward et al., 2003), the above-mentioned limitation is an acceptable compromise.
* * *
**Referee's Comment**

SPECIFIC COMMENTS

-Page 5, Line 28: You should mention what is presented in the figure.

**Authors' Response**

We mentioned what is presented in the figure on the same page (Lines 15 – 18, and 20 – 21) and in the figure caption (Page 6, Lines 1 – 3). We would appreciate more clarity on what the referee meant by the above comment.
* * *
**Authors' Response**

We now cited my Ph.D. dissertation (Santikari, 2017), which describes the study area in more detail than could be presented in the current paper.
* * *
**Authors' Response**

Method of calculation is now mentioned as:

The calculated values of $I_a$ [using eqs. (3), (6), and $\lambda = 0.2$] for watersheds…
* * *
-Please avoid using plural in parameters symbols. For example, in "Iais" I was initially confused if s was for plural or part of the symbol. You may use other explosion such as "Iai values".

**Authors' Response**

We replaced "$I_{ai}$s" with "$I_{ai}$ values" or "values of $I_{ai}$". "$Q_i$s", "CN$_i$s" were also replaced similarly.
* * *
**Referee's Comment**

-Page 9, Lines 9-11: As it is explained in Soulis and Valiantzas (2012), the reason is the non-linear form of the SCS-CN formula. So, the average of the results is not equal with the result using average value of the parameters.

**Authors' Response**

It is true that non-linear form causes the runoff from averaging CN to be different than the runoff from averaging $Q$. However, the purpose of lines 9-11 is to mention that averaging $Q$ is a better approach than averaging CN, rather than to explain why they give different results.
* * *
**Referee's Comment**

-Page 9, Line 13: "to be"

**Authors' Response**

Grammatical error corrected
* * *
**Referee's Comment**

-Page 9, "2.2. Ia in a Heterogeneous Watershed": It should be mentioned that the following justification is valid in the case that each subarea is directly connected with the drainage network. This is a logical assumption in most cases, especially when there is a dense drainage network, however, it is still an assumption.

**Authors' Response**

The referee makes a good point. However, we would like to point out that all CN models inherently make this assumption when the watershed is discretized. There is no provision in the CN method to handle overland flow crossing subarea boundaries. More process based models, e.g. GSSHA (Downer and Ogden, 2004), may be required to model such interactions between the subareas. In any case, for more clarity, we modified the sentence at page 9, line 19 as:

If each land use type is assumed to be directly connected to the drainage network, the number of land use types contributing to the runoff, in other words the runoff contributing area, increases with rainfall.
* * *
**Referee's Comment**

-Page 34, Lines 9-14: You could use additional evaluation criteria e.g. the relative NSE (rNSE) and the NSE with logarithmic values (lnNSE) to reduce the problem of the NSE sensitivity to extreme values (see Krause et al., 2005).

**Authors' Response**

We thank the referee for this information. It is impossible to calculate lnNSE because the conventional models predicted zero-runoffs in small events (page 24, lines 23 – 25). However, we calculated rNSE and added the results to Table 3 and the newly added Table 4 (see Response to Referee #1). The updated tables are attached at the end of this response. Large storms still had greater influence on rNSE than small storms because the runoff from large storms was an order of magnitude greater than the average runoff.  So the rNSE values are still close to unity but they varied slightly between the models. The

relative performance of the models remains unchanged.
* * *
**Referee's Comment**

-"Model Suitability" section: It would be interesting if you could at least discuss (if it is not possible to compare) with the Soulis and Valiantzas 2012 and 2013 methods, which provided the base for this study.

**Authors' Response**

As we mentioned (page 7, lines 3-7), the methods used by Soulis and Valiantzas (2012, 2013) are equivalent to a distributed parameter CN model. We described in a response above that we added a paragraph under "6.4. Model Suitability" to discuss modified models in comparison with distributed parameter models.
* * *
Conclusively, based on the above comments, I believe that this paper is really interesting and worth being published in case that the authors are able to address the above issues.

* * *
RFERENCES

[revised manuscript text omitted]

---

## Author Response (AR3)

We appreciate the input received from the referees and the editor during the review process, and this has improved the quality of our manuscript. We have made the minor corrections suggested by Referee#2 as described below.

**Referee's Comment**

I believe that the revised manuscript was substantially improved and the authors answered/addressed adequately almost all of the reviewers' comments. The main weakness of the manuscript (i.e. strong reliance on a reference to an unpublished paper) was more or less addressed with the addition of a citation to the Ph.D. dissertation (Santikari, 2017), which is available online. I read the dissertation (at least partly) and it covers the issues raised by the reviewers, so, it is ok with me. However, the editor may finally suggest if it is allowed to add a citation to an unpublished paper or if only the citation to the Ph.D. dissertation should remain. Though, I believe that the objectives of this study (Page 2, lines 15-19) should be revised in order to also include the citation to the Ph.D. dissertation of Santikari, (2017).

**Authors' Response**

The objectives have been revised to include citations to the Ph.D. dissertation as follows:

The objective of this work is to improve the event-scale lumped-parameter application of the CN method by describing an approach for incorporating the spatiotemporal variations of CN. The investigation is described in two papers, which build on Santikari (2017). In this paper, effects of spatial variation of CN (heterogeneity) at the watershed scale are analyzed. Insights gained from this analysis are used to create modified models that account for heterogeneity. The modified models are evaluated using the runoff generated by a distributed parameter model applied to a hypothetical heterogeneous watershed. In a companion paper (Santikari and Murdoch, 2018) and in Santikari (2017), the modified models are refined by including an approach that accounts for the temporal variation of CN using antecedent moisture. The refined models, which account for spatial and temporal variability, are then evaluated using data from real watersheds.

**Referee's Comment**

Apart from the above, I have only two remaining minor comments.

1. Page 4, Line 9. "If a watershed has multiple land uses or soil types, typically, the CN is areally averaged." I believe that it should be made clear that this happens but it is erroneous in order to avoid confusion to the readers on which is the typical method to estimate runoff in heterogeneous watersheds. Alternatively, this phrase can be removed. Also, I think that "spatially" is more correct than "areally".

**Authors' Response**

To avoid misleading the readers, the sentence has been removed.

**Referee's Comment**

2. Page 5, Lines 4 -14. The definition of the various characterizations of the watersheds according to the relationship between P and CN is a key work presented by Hawkins (1993). Therefore, I believe that the citation of the study of D'Asaro and Grillone (2012) should be removed from this point. This citation was already added to the previous phrase indicating that many researchers have noticed this behavior. Please also check the syntax in lines 5-6 because it is confusing.

Conclusively, I believe that this manuscript is suitable for publication after very minor revisions.

**Authors' Response**

Both citations were removed as both appear in the previous sentence. To avoid confusing syntax, the sentence was shortened as:

In 75% of the watersheds, CN decreased with increasing P and asymptotically approached a constant value.

Citations to Hawkins (1993) that appear later in the same paragraph indicate the importance of his work.

[revised manuscript text omitted]

$$\lambda_{W}